# Ran-GTP assembles a specialized spindle structure for accurate chromosome segregation in medaka early embryos

Ai Kiyomitsu [1], Toshiya Nishimura[2,3], Shiang Jyi Hwang[1], Satoshi Ansai [4,5], Masato T. Kanemaki [6,7], Minoru Tanaka [2] & Tomomi Kiyomitsu [1] ✉

Despite drastic cellular changes during cleavage, a mitotic spindle assembles in each blastomere to accurately segregate duplicated chromosomes. Mechanisms of mitotic spindle assembly have been extensively studied using small somatic cells. However, mechanisms of spindle assembly in large vertebrate embryos remain little understood. Here, we establish functional assay systems in medaka (*Oryzias latipes*) embryos by combining CRISPR knock-in with auxin-inducible degron technology. Live imaging reveals several unexpected features of microtubule organization and centrosome positioning that achieve rapid, accurate cleavage. Importantly, Ran-GTP assembles a dense microtubule network at the metaphase spindle center that is essential for chromosome segregation in early embryos. This unique spindle structure is remodeled into a typical short, somatic-like spindle after blastula stages, when Ran-GTP becomes dispensable for chromosome segregation. We propose that despite the presence of centrosomes, the chromosome-derived Ran-GTP pathway has essential roles in functional spindle assembly in large, rapidly dividing vertebrate early embryos, similar to acentrosomal spindle assembly in oocytes.

During early embryogenesis in animals, a large, fertilized egg undergoes repeated cell divisions to create numerous small, differentiated cells[1]. This process comprises a series of dynamic physical and biochemical changes, including cell size reduction[2–7], zygotic gene activation[8–11], and cell cycle remodeling[12,13]. Regardless of these drastic cellular changes, unified parental chromosomes must be accurately duplicated and segregated to all blastomeres to maintain and transmit genomic information. Although sizes and cleavage patterns of fertilized eggs vary among species[14], a microtubule-based bipolar structure, the mitotic spindle[15], is generally assembled in each blastomere to segregate duplicated chromosomes into daughter cells. Recent

studies have shown that embryonic divisions in bovines and humans are error-prone[16,17], but mechanisms of spindle assembly and chromosome segregation in vertebrate embryos remain poorly understood, compared to those in somatic cells.

In the 1980s, based on dynamic properties of microtubules (MTs), Kirschner and Mitchison proposed a simplified search and capture model[18,19] for mitotic spindle assembly. In this model, centrosomes act as the major MT organizing center, and dynamic MT plus-ends are captured by kinetochores on chromosomes to form a bipolar spindle. On the other hand, in the 1990s, Heald and colleagues proposed a self-organization model using *Xenopus* egg extracts[20,21]. In this model,

[1]Okinawa Institute of Science and Technology Graduate University, 1919-1 Tancha, Onna-son, Kunigami-gun, Okinawa 904-0495, Japan. [2]Division of Biological Science, Graduate School of Science, Nagoya University, Chikusa-ku, Nagoya 464-8602, Japan. [3]Hokkaido University Fisheries Sciences, 3-1-1, Minato-cho, Hakodate, Hokkaido 041-8611, Japan. [4]Graduate School of Life Sciences, Tohoku University, Sendai, Miyagi 980-8577, Japan. [5]Laboratory of Genome Editing Breeding, Graduate School of Agriculture, Kyoto University, Sakyo-ku, Kyoto 606-8507, Japan. [6]Department of Chromosome Science, National Institute of Genetics, Research Organization of Information and Systems (ROIS), and Graduate Institute for Advanced Studies, SOKENDAI, Yata 1111, Mishima, Shizuoka 411-8540, Japan. [7]Department of Biological Science, The University of Tokyo, Tokyo 113-0033, Japan. ✉e-mail: tomomi.kiyomitsu@oist.jp

chromatin acts as a MT nucleation site, and MTs around chromosomes subsequently coalesce and focus into spindle poles in the absence of centrosomes. Afterward, several studies established the key concept that conserved chromatin-bound RCC1 (regulator of chromosome condensation 1) generates a spatial Ran-GTP gradient[22–27], which promotes self-organization of the spindle by locally activating spindle assembly factors (SAFs) near chromosomes[28–30]. Consistent with this model, Ran-GTP is essential for acentrosomal spindle assembly in female meiosis[31–33], with more functional significance in meiosis II in mice[31]. However, we recently found that whereas Ran-GTP affects localization of some of SAFs, the Ran-GTP pathway is dispensable for bipolar spindle assembly in centrosome-containing somatic human cells[34], as observed in other somatic cell lines[35,36], probably due to functions of centrosomes and other pathways.

What function does Ran-GTP serve in spindle assembly in large, centrosome-containing vertebrate embryonic cells? Considering the large distance between chromosomes and centrosomes (>20 μm in *Xenopus laevis* early embryos[4,37]), Ran-GTP may have unique, essential functions for embryonic spindle assembly, despite the presence of centrosomes. Interestingly, in *Xenopus laevis* embryonic extracts, Ran-GTP is required for spindle assembly at the 4-cell stage, but not at the ~4000 cell stage[3]. However, it remains unclear how Ran-GTP promotes spindle assembly and chromosome segregation in live *Xenopus* embryos and in other species. In addition, even though mechanisms of spindle scaling, including the upper limit to mitotic spindle length have been well investigated[3,4,6,7,38], it remains unclear whether and how early embryos assemble a specialized spindle structure for accurate chromosome segregation during cleavage.

Fish embryos generate large, centrosome-containing spindles[7,39–41]. Importantly, early divisions are planar[2,14,40]. They occur synchronously in a relatively uniform, single cell-layer sheet at the animal pole, providing an ideal opportunity to observe dynamic processes of spindle assembly and positioning in all blastomeres from 1-cell to 16-cell stages. Compared to zebrafish, Japanese medaka, *Oryzias latipes*, has several advantages, including smaller genome size[42], a wide range of permissive temperatures, and daily egg production[43], which are helpful for genome editing and handling of embryos. In this study, we analyzed dynamic mechanisms of spindle assembly and chromosome segregation in medaka early embryos by combining high-quality live imaging with CRISPR/Cas9-mediated genome editing and an auxin-inducible degron 2 (AID2)-based protein knockdown system[44]. Our live functional studies revealed that Ran-GTP is required for accurate chromosome segregation because it assembles a dense MT network around the metaphase spindle midplane, specifically in early embryos.

## Results

### Live imaging of chromosomes and MTs in medaka embryos
Medaka early embryonic development was carefully observed using a stereo microscope[2]. To investigate intracellular dynamics, we first generated transgenic medaka expressing RCC1-mCherry2 (RCC1-mCh) and EGFP-α-tubulin and visualized chromosomes and MTs using a spinning-disc confocal microscope (Fig. 1a, b). The mCherry2 coding sequence (CDS) was integrated into the C-terminal region of the endogenous RCC1 gene using 5′-modified dsDNA donors[45] (Supplementary Fig. 1a), whereas CDS of EGFP-α-tubulin was inserted at the 5′ UTR of the tubulin alpha-1B gene on Chromosome 7 (Supplementary Fig. 1b) (see Methods for details). Homozygous adult medaka having both RCC1-mCh and EGFP-α-tubulin grew normally, and fertilized eggs obtained from homozygous pairs were used for live imaging (Fig. 1b).

RCC1 is a well-conserved chromatin-binding protein[46–48] (Supplementary Fig. 1c). RCC1-mCh co-localized with Histone H2B (a marker of chromosomes) (Fig. 1c) and visualized both Meiosis II chromosomes and polar bodies (PBs) (Fig. 1d). After fertilization, a female pronucleus

migrated (Fig. 1b t = 0) toward a male pronucleus which had centrosomes, located around the center of the blastodisc near the animal pole (Fig. 1a, b). In contrast, oil droplets were displaced from the animal pole to the vegetal pole before the first mitosis[2] (Fig. 1b). This oil droplet migration is well known, but the driving force is unclear. Interestingly, comet-like structures of EGFP-α-tubulin were formed under these oil droplets (Fig. 1e), which contained microtubule-like signals (Supplementary Fig. 1d) and elongated radially during oil droplet displacement (Fig. 1b, t = 0–18). These results show that dual-color, live imaging is a powerful method to reveal uncharacterized intracellular dynamics in embryogenesis.

### Characterization of medaka cleavage
To characterize spindle assembly and cell cycle remodeling during early embryogenesis, we next performed long-term, live-cell imaging of fertilized medaka eggs at 3-min intervals for ~10 hr at 24–25 °C. Metaphase spindles were visualized from the 1-cell to late blastula stage (Fig. 2a, Supplementary Fig. 2a, and Supplementary Movie 1). Despite long-term imaging, 82% (n = 18/22) of embryos developed normally and hatched like control embryos without imaging (85%, n = 28/33), suggesting very low phototoxicity of these imaging conditions.

We first measured mitotic duration from nuclear envelope breakdown (NEBD) to anaphase onset. In contrast to the longer duration in mammalian zygotes (> 60 min in mice[5], bovines[16], and humans[17]), the first division in medaka required only ~12 min and the duration decreased further to ~9 min at the 256-cell stage (Fig. 2b). Despite the short duration, chromosome segregation errors such as anaphase lagging chromosomes were never observed during the first 4 divisions (n = 18, Supplementary Movie 1). The cell cycle length (duration between successive metaphases) was ~36 min from the first to the second division, and the length was constantly ~30 min until the 9th division (256-cell stage) (Fig. 2c), which is similar to the duration in zebrafish (~15 min) and *Xenopus* (~30 min), but shorter than that in mice (~12 hr)[12]. The inner blastomeres of 8-cell stage embryos, which are smaller than outer blastomeres (Supplementary Fig. 2b), appeared to lengthen the cell cycle at the ~512-1024 cell stage (10-11th divisions), one round earlier than peripheral blastomeres (Fig. 2c). The timing of this cell-cycle increase seems to be roughly coupled with the timing of zygotic gene activation[8] (Supplementary Fig. 2a).

Zygotic spindles were oriented horizontally to achieve planar divisions until the 16-cell stage, but spindles in the central four blastomeres or other relatively smaller cells tended to orient perpendicularly during the 5th division (16-cell stage metaphase, Fig. 2a, t = 123). Anaphase entry was well synchronous until the 16-cell stage, but some smaller cells entered anaphase earlier, at around the 32-cell stage (Fig. 2a, t = 150), consistent with a previous study[49]. Many cells were not synchronous around the late morula stage (256–512 cell stage, Fig. 2a, t = 231), after which blastomeres started to migrate in concert with cell cycle elongation (Fig. 2c, Supplementary Fig. 2a and Movie 2). Although abnormal divisions were rarely observed until the early morula (64–128 cell) stage, late blastula embryos displayed several larger polyploid-like nuclei (Supplementary Fig. 2c, n = 25 embryos). These large nuclei seemed to be generated by previously uncharacterized cell fusion during the late morula stage (Supplementary Fig. 2d, Movie 2). Together, these results provide fundamental information about cell cycles in medaka embryology.

### Centrosome positioning in medaka early embryonic divisions
Mitotic spindle position and orientation are determined by the position of interphase centrosomes in early frog and fish embryos[39]. Since spindle orientation is well regulated during the first 4 divisions in medaka embryos (Fig. 2a), we next analyzed centrosome positioning. Centrosomes, marked by punctate tubulin signals[39], were always positioned at opposite sides of the nucleus before mitotic entry

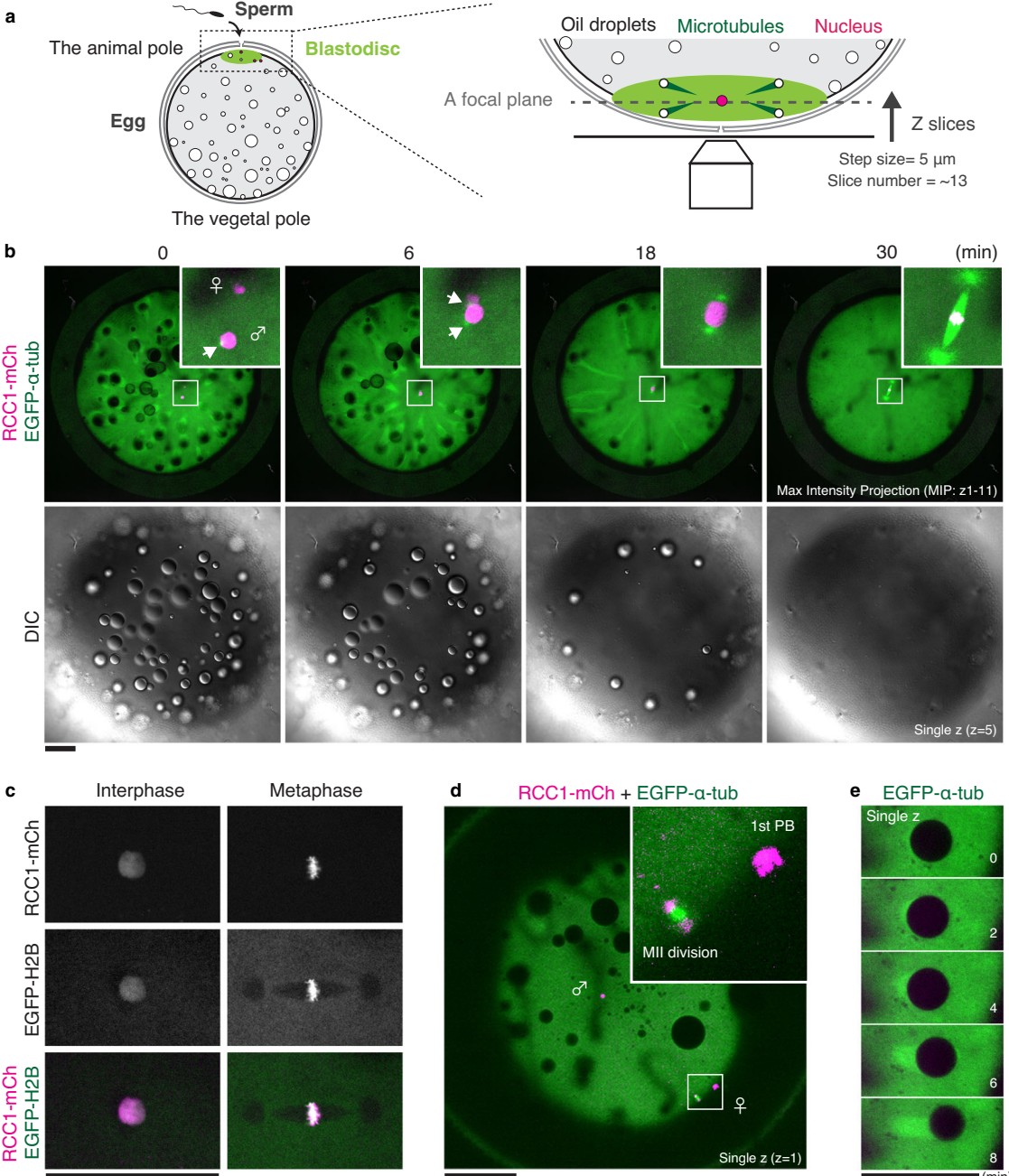

**Fig. 1 | Live imaging reveals dynamic behavior of chromosomes and MTs in medaka fertilized eggs. a** Schematic representation of microscopic observation of a medaka fertilized egg. **b** Live-embryo images showing dynamics of chromosomes (magenta), MTs (green), and oil droplets. Maximum-intensity-projection (MIP) images are shown. Insets indicate pronuclear fusion and a zygotic spindle. White arrows indicate centrosomes. **c** Live-cell fluorescent images showing co-localization of RCC1-mCh with EGFP-Histone H2B. **d** A single-plane live-cell fluorescent image showing the first polar-body (PB) and Meiosis II division at the periphery of the blastodisc. **e** Representative GFP-α-tubulin fluorescent images showing generation of a comet-like structure under an oil droplet. Scale bars = 100 μm.

(Fig. 2d t = −12, 24). After chromosome segregation, centrosomes were duplicated in the cytoplasm and separated vertically relative to the spindle elongation axis (Fig. 2e t = 4, 2 f, Type I), consistent with a previous study[39]. Daughter nuclei subsequently migrated between the separated centrosomes (Fig. 2e t = 8) before the next mitosis. In some cases, however, centrosomes were transiently separated horizontally (Fig. 2e–g Type II, t = 4), but finally became oriented vertically to the spindle elongation axis (Fig. 2e, f, Type II). The frequency of Type II orientation was relatively high in the first division, but reached ~50% in the fourth divisions (Fig. 2g). The Type I and II configurations were also detected by γ-tubulin, a centrosome marker[41] (Supplementary Fig. 2e), which showed punctate signals at the center of asters in metaphase (Supplementary Fig. 2f). These results suggest that centrosomes are not attached to the nucleus transiently after early embryonic divisions and that their positioning is regulated by multiple mechanisms to ensure proper mitotic spindle orientation.

**Dynamic changes of spindle architecture during embryogenesis**
Since mitotic spindle size scales with cell size in early embryonic divisions[4–7], we next measured metaphase spindles in different stages (Fig. 3a–e). Spindle length, defined as centrosome-centrosome distance (solid lines in Fig. 3a), decreased significantly from ~70 to 10 μm

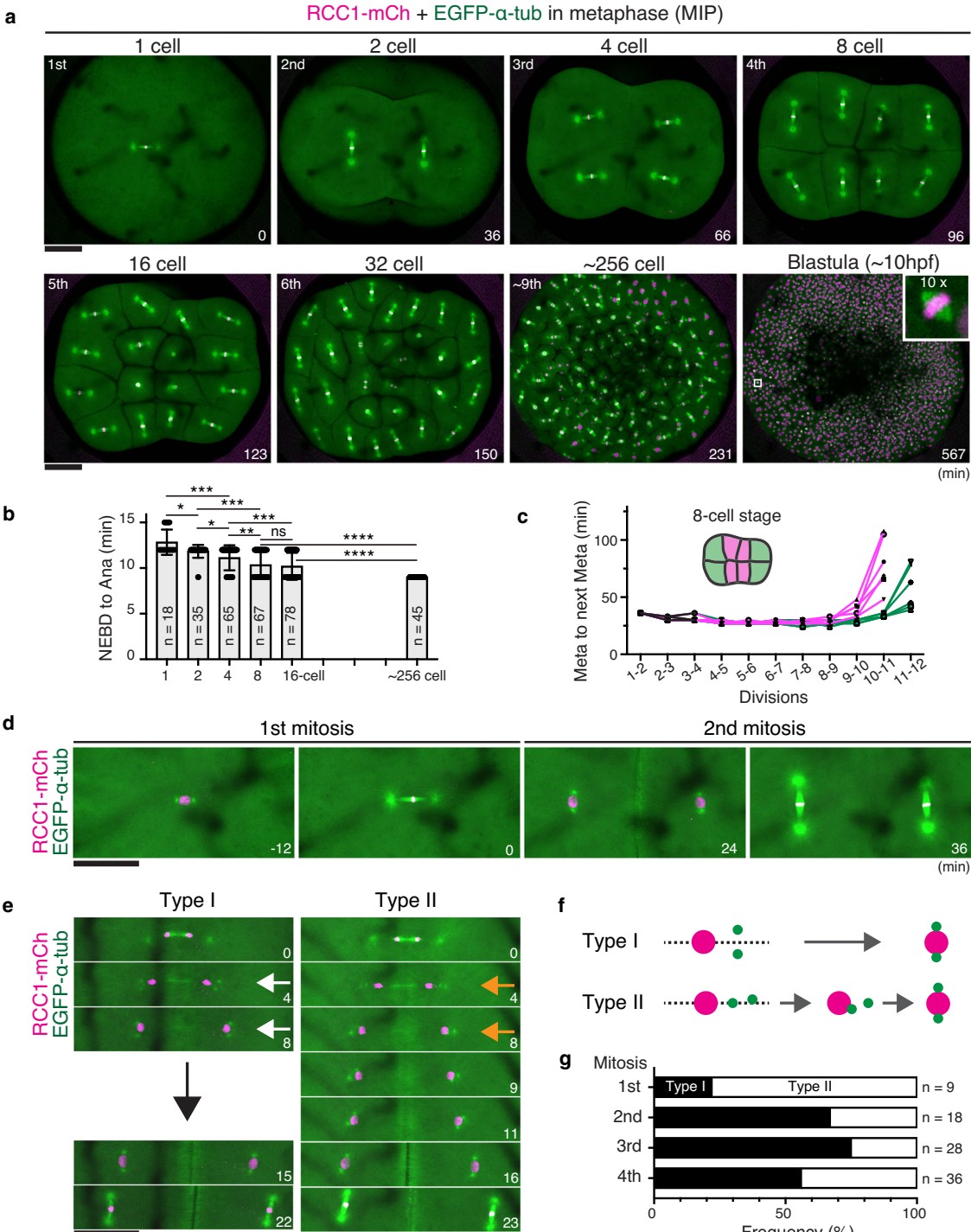

**Fig. 2 | Live imaging shows dynamic regulation of the cell cycle and spindle positioning during medaka early cleavage. a** Representative live-cell images showing metaphase spindles in indicated embryonic stages. **b** Quantification of time elapsed from NEBD to anaphase onset at each stage. Indicated numbers of blastomeres in focal planes were measured in 18 embryos. **c** Quantification of time from one metaphase to the next. Cell-division lineages were tracked and measured for inner (magenta) and peripheral (green) blastomeres, respectively ($n = 6$). **d** Live-cell images showing the correlation between centrosomes and spindle orientation in the first and second mitosis. **e** Centrosomes are initially separated either vertically (Type I, white arrows) or horizontally (Type II, orange arrows) after mitosis. Centrosomes in Type II cells subsequently oriented vertically before entering the next mitosis. **f** Classification of centrosome separation patterns based on (**e**). **g** Quantification of centrosome separation types in early embryonic divisions. Error bars indicate mean ± SD. Scale bars = 100 μm. Source data for (**b**, **c**, **g**) are provided as a Source Data file. One-way ANOVA with Dunnett's multiple comparisons test was performed for (**b**). *$p < 0.1$, **$p < 0.01$, ***$p < 0.001$ and ****$p < 0.0001$.

(Fig. 3c), whereas spindle width and metaphase plate length were relatively constant until the early blastula stage (Fig. 3d and Supplementary 3a). The first spindles were the longest, but thinner and more prone to bending compared to the second spindles (Figs. 2d and 3d, Supplementary Fig. 3b $n = 18$). Spindle length appeared to reach an upper limit in the first to fourth (1-cell to 8-cell) divisions (Fig. 3e), but scaled with cell diameter after the 16-cell stage, as observed in other vertebrates[4,5,7,37].

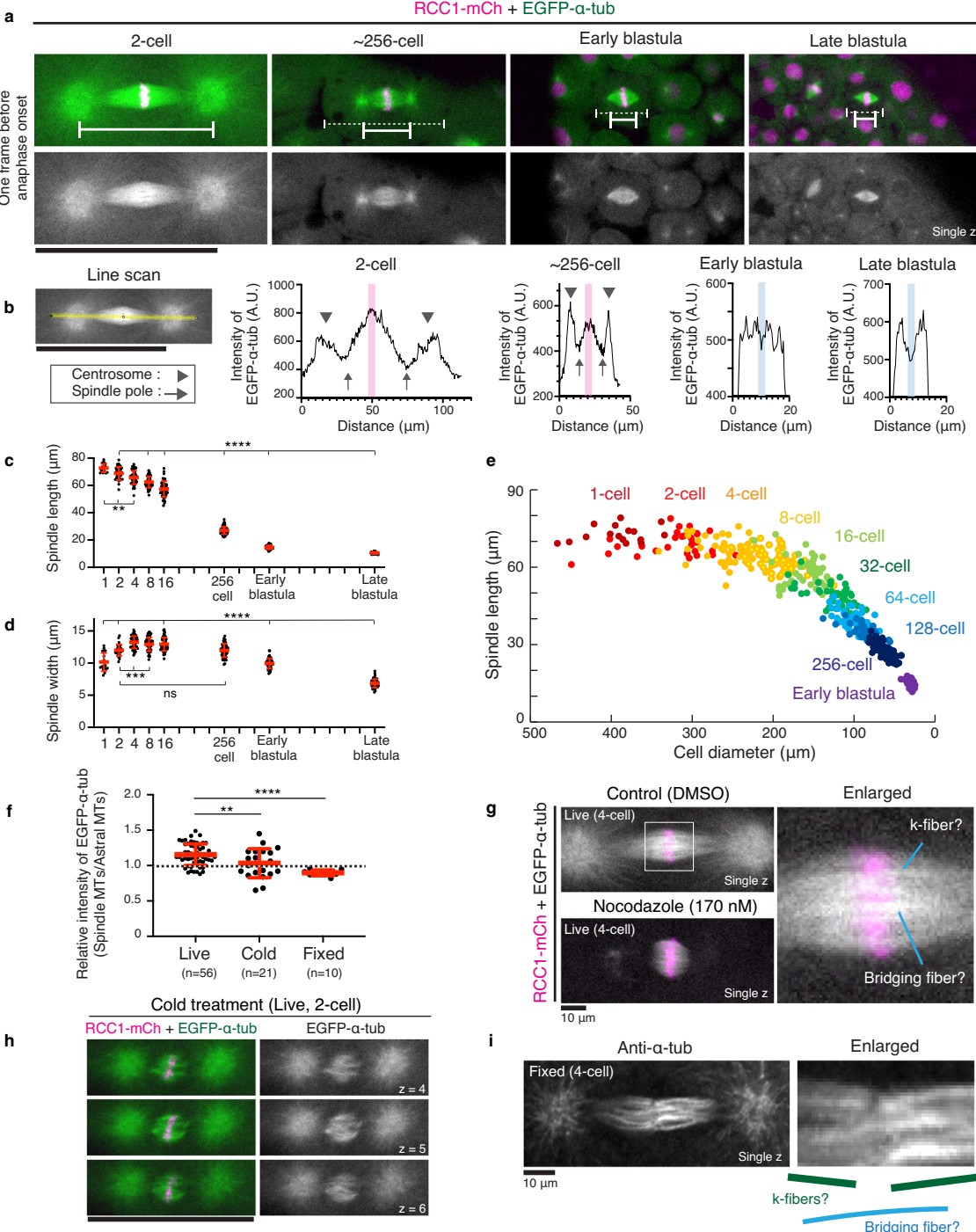

**Fig. 3 | Mitotic spindles display dynamic structural changes during medaka early cleavage. a** Representative live-cell images showing metaphase spindles in indicated embryonic stages. Solid and dashed lines indicate the centrosome-centrosome distance and the cell diameter, respectively. **b** Left: Fluorescence intensity on the yellow line was measured. Right: Graphs of fluorescence intensities for line scans of spindles in (**a**) showing an increase of EGFP-α-tubulin intensity at the spindle midplane in 2-cell and ~256-cell, but not in early or late blastula spindles. **c, d** Quantification of spindle length (**c**) and width (**d**) in indicated stages. $n$ = 18, 31, 50, 54, 50, 87, 64, and 51 for (**c**) and $n$ = 18, 29, 51, 58, 58, 50, 50, and 54 for (**d**) from 1-cell to late blastula stages, respectively, from 18 different embryos. **e** Cell diameter and spindle length are plotted as colored circles for individual embryos at different stages, showing spindle length scales with cell diameters after 16-cell stages. **f** Quantification of the max value of EGFP-α-tubulin intensity on spindle MTs relative to that on astral MTs from 2 or 4 cell spindles. **g** Left: live-cell images of control (top) and nocodazole-treated (bottom) spindles in metaphase. Right: an enlarged image showing k-fiber- and bridging-fiber-like bundled MTs at the spindle center. **h** Live-cell images of a metaphase spindle after cold treatment. **i** An image of a metaphase spindle after fixation showing k-fiber- and bridging-fiber-like bundled MTs at the spindle center. Error bars indicate mean ± SD. Scale bars = 100 μm (**a, h**) and 10 μm (**g, i**). Source data for (**b**–**f**) are provided as a Source Data file. One-way ANOVA with Dunnett's multiple comparisons test was performed for (**c, d, f**). **$p < 0.01$, ***$p < 0.001$ and ****$p < 0.0001$.

Strong astral MTs were clearly visible around centrosomes in early embryonic spindles (Fig. 3a, b). Unexpectedly, early embryonic spindles have another MT-dense region around metaphase chromosomes (Fig. 3b, indicated in magenta), which has higher tubulin intensities relative to those at centrosomes during the 1st to 4th divisions (Fig. 3f, Supplementary Fig. 3c). Both central and astral MTs gradually diminished and became undetectable after early blastula stages (~1024-cell stage, Fig. 3a, b, Supplementary Fig. 3d). These results indicate that not only spindle size, but also its architecture dynamically changes during embryogenesis in medaka.

### The central MT network consists of dynamic and stable MTs

To understand characteristics of spindle MTs in early embryos, we next treated embryos with 170-nM nocodazole, a MT-destabilization drug (Fig. 3g). Although astral MTs were destabilized, the central MT network can be formed even in the presence of nocodazole, suggesting that central MTs are more stable than astral MTs.

To understand stabilities of central spindle MTs, we next treated metaphase embryos with ice-cold solution. At room temperature, multiple bundled MTs that resemble kinetochore-fibers (k-fibers)[15] and bridging fibers[50,51] or midplane-crossing MTs[52] were observed in the dense MT region at the spindle center (Fig. 3g right). These bundled MTs remained after cold treatment (Fig. 3h), but other short, non-bundled spindle MTs appeared to be selectively destabilized, resulting in a slight reduction of MT intensity at the spindle center relative to astral MTs (Fig. 3f). Fixation using 4% PFA and acetone also reduced intensities of central spindle MTs relative to astral MTs (Fig. 3f, Supplementary Fig. 3e). The remaining EGFP-α-tubulin was too dim to visualize spindle MTs, but immuno-fluorescence using anti-α-tubulin antibody produced very bright signals and visualized individual, bundled MTs reminiscent of k-fibers and bridging-fibers (Fig. 3i). These k-fiber-like bundled MTs showed higher intensities near chromosomes (Fig. 3i, Supplementary Fig. 3f).

Together, these results revealed that the dense MT network at the spindle center consists of at least two MT populations: stabilized, bundled MTs and dynamic MTs sensitive to cold-treatment and fixation (see discussion).

### A dense MT network is formed around chromosomes in metaphase

Early embryos need to efficiently assemble the specialized spindle structure within ~10 min (Fig. 2b). To understand the detailed spindle assembly process, we next performed live-cell imaging at 1-min intervals and tracked behaviors of EGFP-α-tubulin. Although the first mitosis is slightly longer ($10.3 \pm 0.7$, $n = 10$), behaviors of MTs during spindle assembly were almost identical between the first 4 divisions (Fig. 4a–c, Supplementary Fig. 4a–d, $9.0 \pm 0.8$ in the 2nd mitosis, $n = 18$). Before NEBD, centrosomes had intense MT signals, but nuclei were also surrounded by dim MT signals (Fig. 4e t = −1). This was also confirmed by immunofluorescence (Supplementary Fig. 4e). Intriguingly, EGFP-α-tubulin signals suddenly increased in the nuclear region during NEBD (Fig. 4c, e, Supplementary Fig. 4d, t = 0), although fluorescent intensities were weak. Similar nuclear accumulation was observed for EGFP-EB1, a marker of growing MT plus-ends[53] (Fig. 4f) and for mCherry-tagged γ-tubulin, a marker of microtubule minus-ends[15] (Fig. 4g), suggesting that MTs are nucleated around chromosomes at NEBD. Subsequently, MT signals increased at both sides of the chromosome mass to form focused spindle poles next to centrosomes (Fig. 4a t = 1–2, Fig. 4c and Supplementary Fig. 4d t = 2). MT intensities were relatively low on chromosomes (t = 2), but gradually increased throughout the spindle during prometaphase (Fig. 4a t = 3–4, Supplementary Fig. 4d t = 4) and finally formed a dense MT region around chromosomes during metaphase (t = 5–8). GFP-EB1 also showed similar behaviors after NEBD (Fig. 4f) and accumulated at the spindle center during

metaphase (t = 4). mCherry-γ-tubulin signals were also detected on spindles at metaphase (Fig. 4g t = 6).

To carefully analyze dim MT signals generated around chromosomes at NEBD, we next observed spindle assembly in the presence of 170-nM nocodazole (Fig. 4h), which inhibits astral MT growth (Fig. 3g). In this condition, EGFP-α-tubulin signals were more clearly detected in the nucleus at NEBD (Fig. 4h t = 0), which subsequently appeared to self-assemble a bipolar spindle structure (t = 1–2) resulting in formation of intense spindle structure around metaphase chromosomes (t = 3).

To test MT nucleation activities of condensed chromosomes, we next performed MT-regrowth assay coupled with cold treatment. MTs were hardly detectable around chromosomes when embryos were treated just after NEBD (Fig. 4i t = 0). After increasing the temperature, MTs became visible around chromosomes within 75–150 sec, and finally formed densely bundled MTs in 300 sec (Fig. 4i, Supplementary Movie 3). However, as centrosomal MTs appeared to expand concentrically and reached the chromosomal region (Supplementary Movie 3), it was unclear whether condensed chromosomes initiated de novo MT nucleation.

Together, these data suggest that MTs can be nucleated around chromosomes at least in the cage-like nuclear structure at NEBD, which can act as seeds to form a dense MT network around chromosomes during metaphase (see the Discussion).

### Anaphase spindles have no obvious elongation

Generally, somatic spindles elongate during anaphase[54]. Unexpectedly, our live imaging also revealed that early embryonic spindles have no obvious anaphase spindle elongation (Fig. 4a t = 9–12, 4b) like acentrosomal spindles in moss[55], whereas the centrosome-centrosome distance increased gradually without any pauses after NEBD (Fig. 4d).

To understand spindle dynamics during anaphase, we quantified distances between chromosomes, spindle poles, and centrosomes during anaphase (Fig. 5a, b). Their positions were determined based on fluorescent intensity profile (Fig. 5c, see Methods). Whereas pole-pole distances did not change significantly, chromosomes separated toward each spindle pole, resulting in a shorter distance between chromosomes and spindle poles during anaphase (Fig. 5a, b). Immunofluorescence with anti-α-tubulin antibodies also indicated the shortening of bundled k-fiber-like MTs during anaphase (Fig. 5d). Although live images showed intense EGFP-α-tubulin signals between separating chromosomes during early anaphase (Fig. 5a, c t = 1, 2), these signals were hardly detectable after fixation by both EGFP-α-tubulin and anti-α-tubulin antibodies (Fig. 5d, Supplementary Fig. 5a), suggesting that MTs between separating chromosomes in early anaphase are sensitive to our fixation protocol. Interestingly, live images of EMTB-3xGFP, which contains the MT-binding domain of ensconsin[56], were similar to fixed spindle images by anti-α-tubulin antibodies (Fig. 5e, Supplementary Fig. 5b, c), suggesting that EMTB-3xGFP preferentially recognizes fixation-resistant stable MTs. In contrast, EGFP-EB1 accumulated at inter-chromosomal region during anaphase (Fig. 5f).

These results show that anaphase spindles consist of multiple MTs with different stabilities in medaka early embryos. These spindles likely segregate chromosomes in coordination with k-fiber depolymerization as observed in normal somatic cells, but may combine different mechanisms to achieve faster anaphase chromosome movement (Fig. 5b, ~ 3–4 μm/min $n = 8$, see the Discussion).

### No functional spindle assembly checkpoint in early embryos

To further characterize embryonic divisions, we next analyzed anaphase spindle behaviors in nocodazole-treated embryos. 170-nM nocodazole treatment created short barrel-shaped spindles without robust astral MTs (Fig. 3g). However, these spindles entered

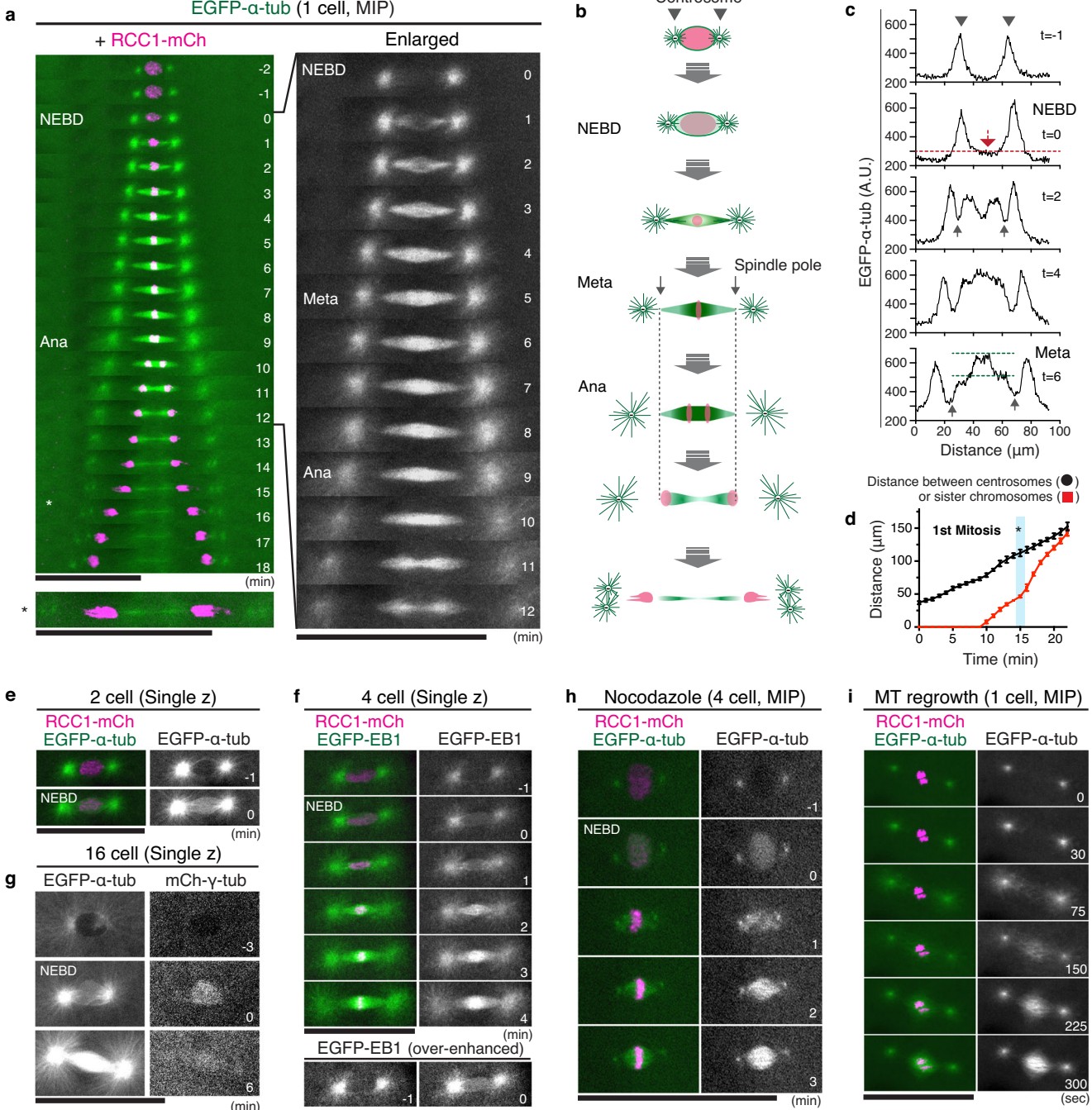

**Fig. 4 | A dense MT network is assembled at the spindle midplane during metaphase in medaka early embryos. a** Kymographs showing the first mitotic spindle assembly process in a medaka embryo. Asterisks indicate a telophase chromosome or deformed nucleus migrating to centrosomes. **b** Schematic representation of spindle assembly and chromosome segregation processes in (**a**). Spindle length is almost constant during anaphase. **c** Graphs of relative fluorescence intensities for line scans of spindles in (**a**). Single z-section images shown in Supplementary Fig. 4d were used for quantification. **d** Quantification of centrosome-centrosome (black) and separating chromosome (red) distance during the first mitosis (*n* = 4 at each time point from 4 embryos). An asterisk shows the timing in (**a**). **e–g** Live-cell images showing nuclear accumulation of EGFP-α-tubulin (**e**, **g**), EGFP-EB1 (**f**), and mCherry-γ-tubulin (**g**) at NEBD (t = 0). **h**, **i** Live-cell images of EGFP-α-tubulin and RCC1-mCherry in the presence of 170-nM nocodazole (**h**) or after MT depolymerization by cold treatment (**i**), showing self-organization of spindle around chromosomes. Error bars indicate mean ± SD. Scale bars = 100 μm. Source data for (**c**, **d**) are provided as a Source Data file.

anaphase with normal timing (Fig. 5g, h), although spindle position and orientation were dysregulated at the 4-cell stage (Fig. 5g). Treatment with 330 nM nocodazole further destabilized MTs (Fig. 5i t = 9), which impaired metaphase chromosome alignment and anaphase chromosome segregation (Fig. 5i t = 12, 24). However, mitotic delay was not observed (Fig. 5h). Intriguingly, during anaphase in 170 nM nocodazole-treated embryos, pole-pole distances slightly

increased (Fig. 5k, l t = 1–2) with bright MT regions separating toward poles in concert with chromosome segregation (Fig. 5k t = 2). Together, these results suggest that large embryonic spindles have no spindle assembly checkpoint and separate chromosomes by multiple mechanisms, including sliding of bridging fibers and k-fiber depolymerization during early anaphase (Fig. 5m, see the Discussion).

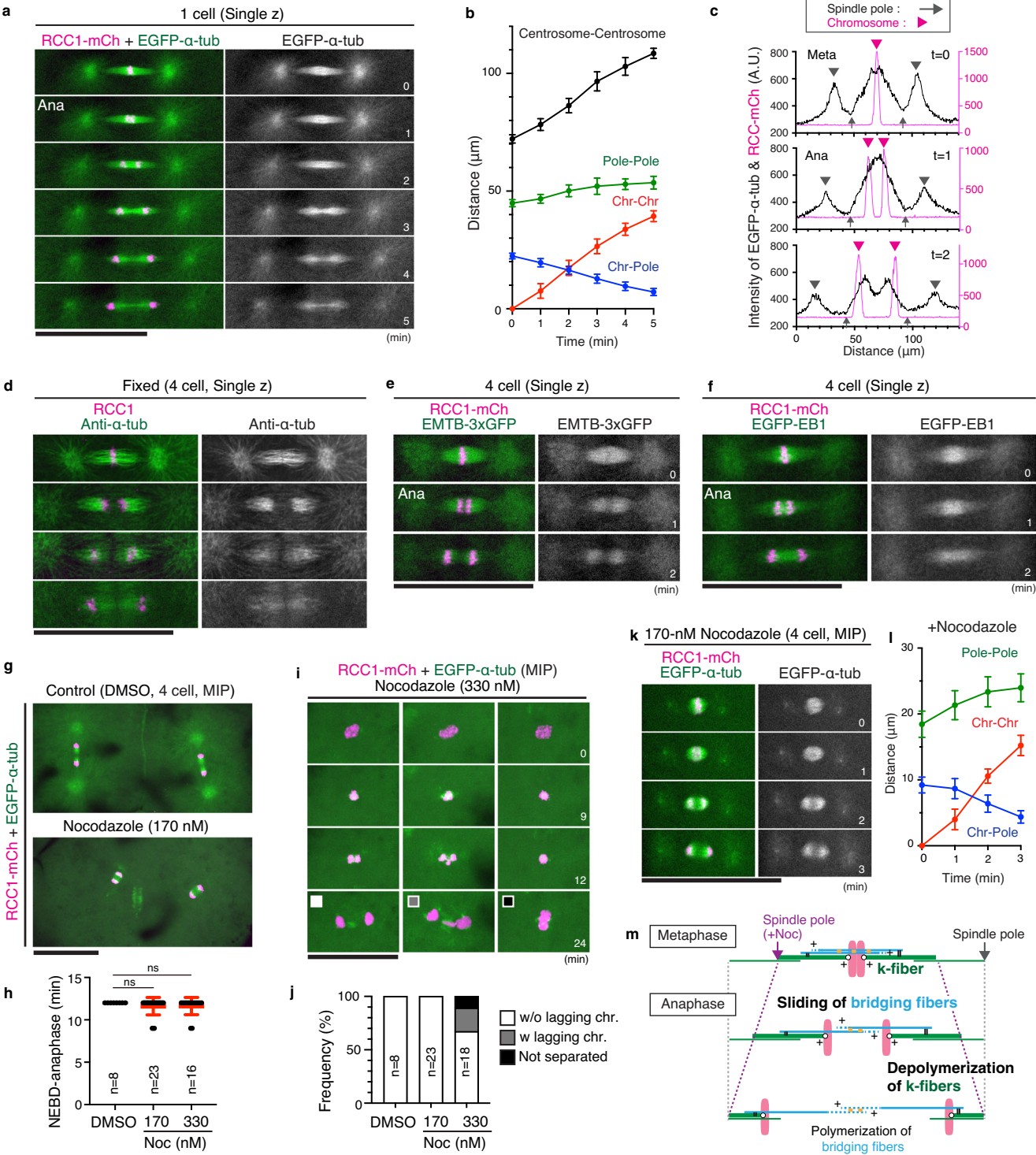

**Fig. 5 | Chromosomes are separated with no obvious anaphase spindle elongation. a** Live-cell images of EGFP-α-tubulin and RCC1-mCherry from metaphase to anaphase. **b** Graphs showing changes of distances between centrosomes (black, *n* = 8), spindle poles (green, *n* = 8), separating chromosomes (red, *n* = 8), and chromosome and spindle pole (blue, *n* = 16) in (**a**) from 8 embryos. **c** Graphs of fluorescence intensities for line scans of spindles in (**a**) showing that EGFP-α-tubulin intensities remain between separating chromosomes during anaphase. The positions of chromosomes (magenta arrowhead), spindle poles (gray arrow), and centrosomes (gray arrowhead) were defined based on the fluorescent intensity profile along the long axis of the spindle. **d** Immunofluorescence images using anti-α-tubulin antibody showing metaphase (top) and anaphase spindles. **e**, **f** Live-cell images of EMTB-3xGFP (**e**) or EGFP-EB1 (**f**) from metaphase to anaphase. **g** Live-cell images of control (top) and nocodazole-treated (bottom) 4-cell spindles in anaphase. **h** Graphs showing the duration of mitosis in DMSO or nocodazole-treated 4-cell blastomeres. **i** 330 nM nocodazole-treated 4-cell blastomeres showing fewer MTs around metaphase chromosomes and abnormal chromosome segregation. **j** Quantification of abnormal segregation phenotypes. **k** Live-cell images of EGFP-α-tubulin and RCC1-mCherry in the presence of 170-nM nocodazole. **l** Graphs showing changes of distances between spindle poles (green, *n* = 11), separating chromosomes (red, *n* = 11), and chromosome and spindle pole (blue, *n* = 22) in (**k**) from 11 embryos. **m** A model of anaphase spindle MTs showing chromosome segregation without spindle elongation. Error bars indicate mean ± SD. Scale bars = 100 μm. Source data for (**b**, **c**, **h**, **j**, **l**) are provided as a Source Data file.

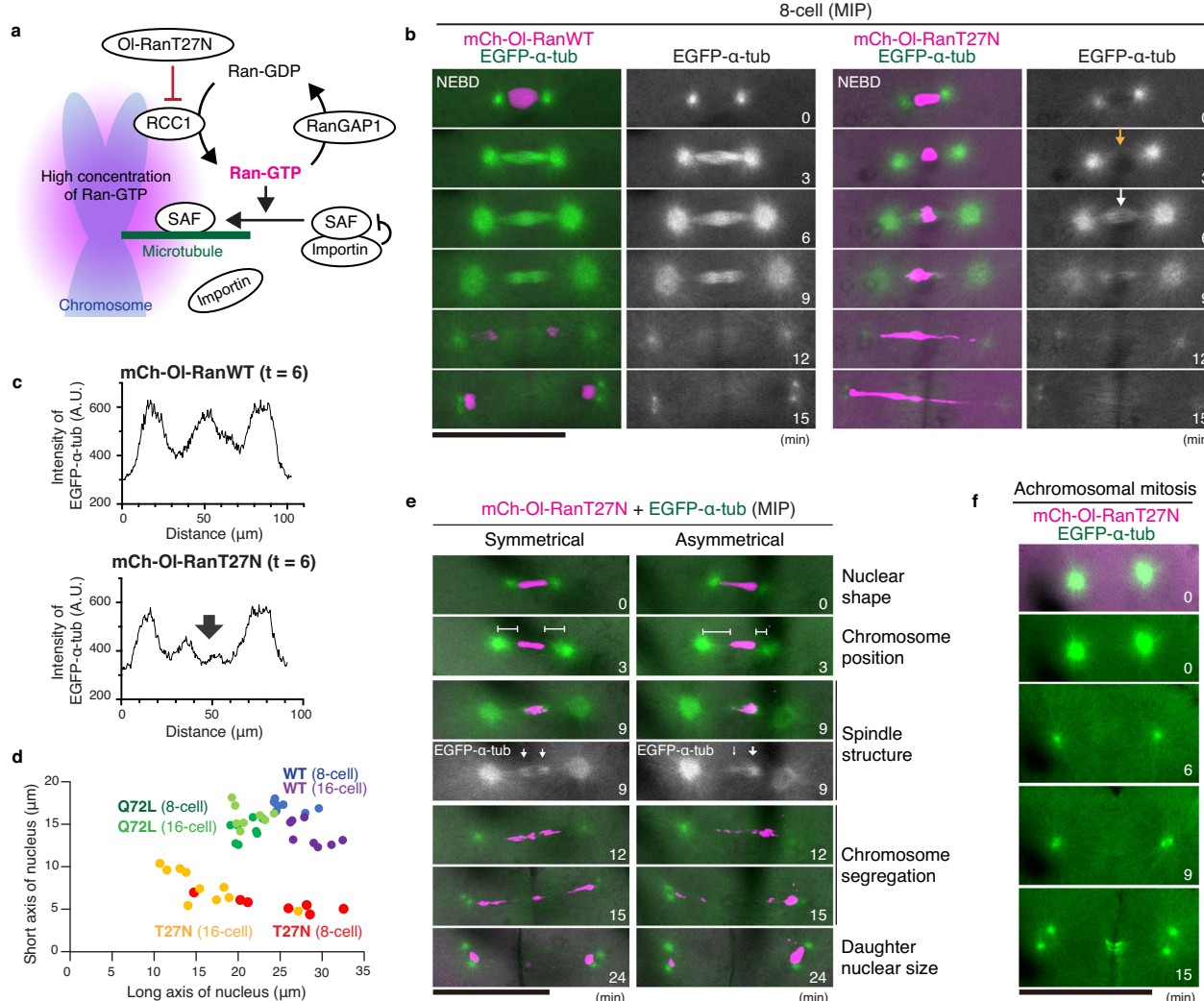

**Fig. 6 | Expression of Ol-RanT27N causes spindle assembly defects followed by abnormal chromosome segregation. a** Schematic representation of Ran-GTP-mediated spindle assembly by activated SAFs. **b** Live-cell images of 8-cell stage embryos showing normal spindle assembly in control (left) and abnormal spindle formation followed by severe chromosome mis-segregation in Ol-RanT27N-expressing embryos (right). **c** Graphs of fluorescence intensities for line scans of Ol-RanWT- (top) or Ol-RanT27N (bottom)-expressing metaphase spindles shown in (**b**). An arrow in (**c**) indicates the decrease of MT intensity at the spindle midplane.

**d** Scatter plots showing nuclear size in 8-cell or 16 cell stage blastomeres expressing Ol-RanWT, Ol-RanT27N, or Ol-RanQ72L mutants. **e** Live-cell images of Ol-RanT27N-expressing embryos showing symmetrical (left) or asymmetrical (right) nuclear shape at NEBD ($t = 0$), which lead to symmetrical or asymmetrical spindle structure ($t = 9$) and chromosome segregation ($t = 12–15$), respectively. **f** Achromosomal mitosis showing no spindle formation between centrosomes. Scale bars = 100 μm. Source data for (**c**, **d**) are provided as a Source Data file.

## Ol-RanT27N expression causes abnormal chromosome segregation

A Ran-GTP gradient has been thought to locally activate SAFs by dissociating inhibitory importins from SAFs around chromosomes[28–30,34] (Fig. 6a). To understand the requirement of Ran-GTP for embryonic spindle assembly in medaka, we next expressed the dominant negative mutant of medaka *Oryzias latipes* Ran, Ol-RanT27N, which corresponds to mouse and human RanT24N (Supplementary Fig. 6a). Exogenous expression of mCherry-tagged wild-type Ol-Ran (mCh-Ol-RanWT) had no obvious effects on spindle assembly at the 8-cell stage (Fig. 6b and Supplementary Fig. 6b). In contrast, expression of the Ran mutant, which binds endogenous RCC1 and inhibits RCC1's GEF activity[57], caused severe chromosome segregation defects (Fig. 6b and Supplementary Fig. 6b, Movie 4) in an expression-level-dependent manner (Supplementary Fig. 6c). 65% ($n = 13/20$) of Ol-RanT27N-expressing embryos showed abnormal segregation in at least one blastomere before the 256-cell stage. Among them, 6 embryos had higher RanT27N expression and displayed phenotypes in 8–16 cell

stages, including chromosome bridge (Fig. 6b, 76% $n = 25/33$) and anaphase lagging chromosomes (12%, $n = 4/33$). Injection of RanT27N mRNA caused embryonic lethality in all embryos ($n = 20$), whereas Ran-WT injection showed normal development and high hatching rate (78%, $n = 7/9$).

Importantly, Ol-RanT27N expression impaired both spindle assembly timing and spindle architecture. Three min after NEBD, spindle MTs were hardly detectable between centrosomes (Fig. 6b and Supplementary Fig. 6c $t = 3$). Subsequently, a faint spindle structure was assembled, but it lacked the dense MT region around chromosomes (Fig. 6b, c and Supplementary Fig. 6c $t = 6$). Chromosomes indicated by mCh-Ol-RanT27N were abnormally elongated after anaphase and finally pinched or cleaved by a cleavage furrow (Fig. 6b and Supplementary Fig. 6c $t = 15$), similar to the fission yeast *cut* phenotype[58].

Ol-RanT27N expression also caused smaller or elongated nuclei with reduced short axes (Fig. 6b $t = 0$, Fig. 6d), which were not induced by Ol-RanWT or Ol-RanQ72L constitutive active mutant (Fig. 6d,

Supplementary Fig. 6d). Intriguingly, among 17 tractable blastomeres showing cut phenotypes, 88% ($n = 15/17$) had symmetrical nuclear shapes (Fig. 6e left t = 0), but the remaining 12% ($n = 2/17$) showed asymmetrical nuclear shapes at NEBD (Fig. 6e right, t = 0), which resulted in asymmetrical chromosome position relative to centrosomes (t = 3), asymmetrical spindle structure (t = 9), unequal distribution of segregating chromosomes (t = 12–15), and unequal-sized daughter nuclei (t = 24). In subsequent divisions, we additionally observed 3 blastomeres showing asymmetrical spindles, which correlated well with asymmetrical nuclear morphology (Supplementary Fig. 6e). Regardless of these abnormal divisions, blastomeres continued embryonic divisions, likely due to the lack of cell cycle and mitotic checkpoints (Supplementary Movie 4). Abnormal division cycles created blastomeres without chromosomes via unequal partitioning of the whole chromosome mass (Supplementary Fig. 6e t = 15). These achromosomal blastomeres entered the next mitosis, but formed no spindle structure between centrosomes (Fig. 6f, Supplementary Fig. 6 f, $n > 10$), although the remaining two centrosomes were sufficient to form a cleavage furrow between them (Fig. 6f t = 15). In contrast to RanT27N, expression of the RanQ72L mutant did not show obvious defects in spindle assembly (Supplementary Fig. 6d), although RanQ72L appeared to accumulate at the nuclear envelop (Supplementary Fig. 6g), and 88% ($n = 15/17$) of mRNA injected embryos did not hatch.

In summary, these results indicate that the chromosome-derived Ran pathway is required to organize proper nuclear structure before entering mitosis and to assemble a dense MT network at the spindle center during metaphase for accurate chromosome segregation in rapid embryonic division cycles.

## RCC1 knockdown causes abnormal chromosome segregation

RanT24N interacts not only with RCC1, but also with importin-β, implying that RanT24N does not act solely as an RCC1 inhibitor[33]. To further validate our results, we next sought to deplete Ran-GTP by degrading RCC1 using auxin inducible degron 2 (AID2) technology[44] (Fig. 7a). Using the same method for mCherry knock-in, we integrated an mAID-mClover-3xFLAG (mACF) coding sequence into the genome in the C-terminal region of the RCC1 gene (Supplementary Fig. 7a). Homozygous knock-in embryos grew normally.

To perform AID2-mediated protein knockdown, we collected naturally fertilized eggs from homozygous knock-in pairs and injected mRNAs encoding OsTIR1(F74G)-P2A-mCherry-Histone H2B (mCh-H2B) into one-cell stage embryos (Fig. 7b, see Methods). mCh-H2B was used as an indicator of OsTIR1 expression as well as a marker of chromosomes. RCC1-mACF signals were visible in control embryos (Fig. 7c), but decreased in response to expression of mCh-H2B in 5-Ph-IAA treated embryos (Fig. 7d), confirming efficient inducible degradation of RCC1 (Supplementary Movie 5). Although mCh-H2B expression levels were slightly variable among embryos, RCC1-mACF fluorescence in the nucleus decreased to ~1.5% of that in controls before the third (4-cell) division (Fig. 7e, $n = 46$ blastomeres from 14 embryos). ~80% of 4-cell blastomeres depleted of RCC1 showed abnormal chromosome segregation phenotypes (Fig. 7f, g), and all embryos ($n = 14$) showed embryonic lethality (Fig. 7h). In the absence of OsTIR1(F74G), 5-Ph-IAA itself had no effects on RCC1 degradation and embryonic development (Supplementary Fig. 7b, all embryos hatched, $n = 7$). In addition, when heterozygous embryos were used, embryos grew normally with high hatching rate (80%, $n = 12/15$), even in the presence of OsTIR1(F74G) and 5Ph-IAA (Supplementary Fig. 7c, d), suggesting that half the normal amount of RCC1 is sufficient for embryonic development.

Homozygous RCC1-depleted blastomeres displayed lagging chromosomes, chromosome bridges, or chromosome non-disjunction during 4-cell divisions (Fig. 7f, g). Phenotypic severity was correlated with the remaining fluorescence intensities of RCC1-mACF (Fig. 7e). Although ~20% ($n = 9/49$) of RCC1-knockdown (KD) blastomeres did

not show obvious chromosome segregation defects at the 4-cell stage (Fig. 7f), all of these daughter cells displayed severe defects in the subsequent 8-cell division (Fig. 7f). As observed in RanT27N expressing blastomeres, nuclei in RCC1-KD blastomeres were relatively smaller and elongated with reduced short axes (Fig. 7g, i). In addition, blastomeres showing chromosome non-disjunction had higher frequencies of asymmetrical nuclear shape, compared to those having the chromosome-bridge phenotype (Fig. 7j, Supplementary Fig. 7e). RCC1-KD blastomeres showed wider metaphase plates (Fig. 7k). These results support our previous results showing that Ran-GTP is required for nuclear formation and chromosome segregation in early medaka embryos.

## RCC1 knockdown disrupts functional spindle assembly

AID2-mediated depletion of RCC1 caused chromosome segregation defects (Fig. 7g). To analyze spindle structure in RCC1 KD embryos, we next expressed OsTIR1(F74G) and mCherry-α-tubulin. As observed in Ol-RanT27N-expressing embryos (Fig. 6b), spindle MTs in RCC1-depleted blastomeres were dimmer relative to astral MTs at prometaphase (Fig. 8a t = 3) and metaphase (t = 6). In addition, RCC1-depleted spindles failed to form the dense MT network around the spindle midplane (Fig. 8a, b and Supplementary Fig. 8a), which showed shorter spindle width (Fig. 8c) and longer spindle length (centrosome-centrosome distance) (Fig. 8d). Consistent with above results (Figs. 6e, 7i, j), RCC1 KD also caused asymmetrical nuclei (Fig. 8e t = 0, $n = 3/16$ from 6 embryos), 67% of which led to chromosome non-disjunction (Fig. 8e, t = 3–12). Our live imaging indicated that asymmetric nuclear formation came from improper attachment of one of two centrosomes (Fig. 8e t = −18), which may be caused more frequently in the Type II nucleus-centrosome attachment pathway (Fig. 2f, Supplementary Fig. 8b).

In RCC1-KD embryos, spindle MTs were still visible around chromosomes (Fig. 8a t = 6). To test stabilities of these spindle MTs, we next treated RCC1-depleted embryos with nocodazole. 330 nM nocodazole was used since effects of nocodazole appeared to decrease after the mRNA injection and 5-Ph-IAA treatment. In RCC1-depleted embryos, nocodazole-stable MTs were still observed, but intensities were reduced (Fig. 8f, g). Together, these results indicate that Ran-GTP is required for proper nucleus-centrosome connection and functional spindle assembly in medaka early embryos.

## RCC1 is dispensable for spindle assembly in blastula-stage

In stark contrast to the present results (Fig. 7g), we previously reported that RCC1 depletion in HCT116 human cells does not result in abnormal chromosome segregation phenotypes[34]. To analyze the requirement of Ran-GTP in later-stage live embryos, we next added 5-Ph-IAA 8 hr post-OsTIR1(F74G) injection (hpi), which corresponds to blastula stage (Supplementary Fig. 2a). In the presence of both OsTIR1(F74G) and 5-Ph-IAA, RCC1-mACF signals were diminished and reached an undetectable level within 45 min (Fig. 9a). To monitor chromosomes with mCh-α-tubulin, H2B fused with miRFP670-Nano3[59] was used, which produces bright signals in later stages (Fig. 9a, b). In contrast to the case in early embryos, RCC1-KD did not cause any significant changes in nuclear size at NEBD in blastula-stage embryos (Fig. 9b, c, t = 0). Bipolar spindles were also assembled (Fig. 9d), but they were slightly shorter (Fig. 9e) and appeared to have less intense microtubules around chromosomes (Fig. 9d, f), as observed in RCC1-depleted cultured human cells[34]. Mitosis was slightly prolonged (Fig. 9g), but abnormal chromosome segregation was rarely observed (Fig. 9h). Instead, RCC1-depleted blastomeres failed to form round nuclei with decondensed chromosomes after mitotic exit (Fig. 9i), which most likely caused lethality of RCC1-depleted embryos (0% hatching rate, $n = 0/22$, compared to 70% in control, $n = 7/10$). These data indicate that Ran-GTP becomes non-essential for functional spindle assembly and chromosome segregation, but remains indispensable for nuclear envelope reformation after mitotic exit in later-stage embryos.

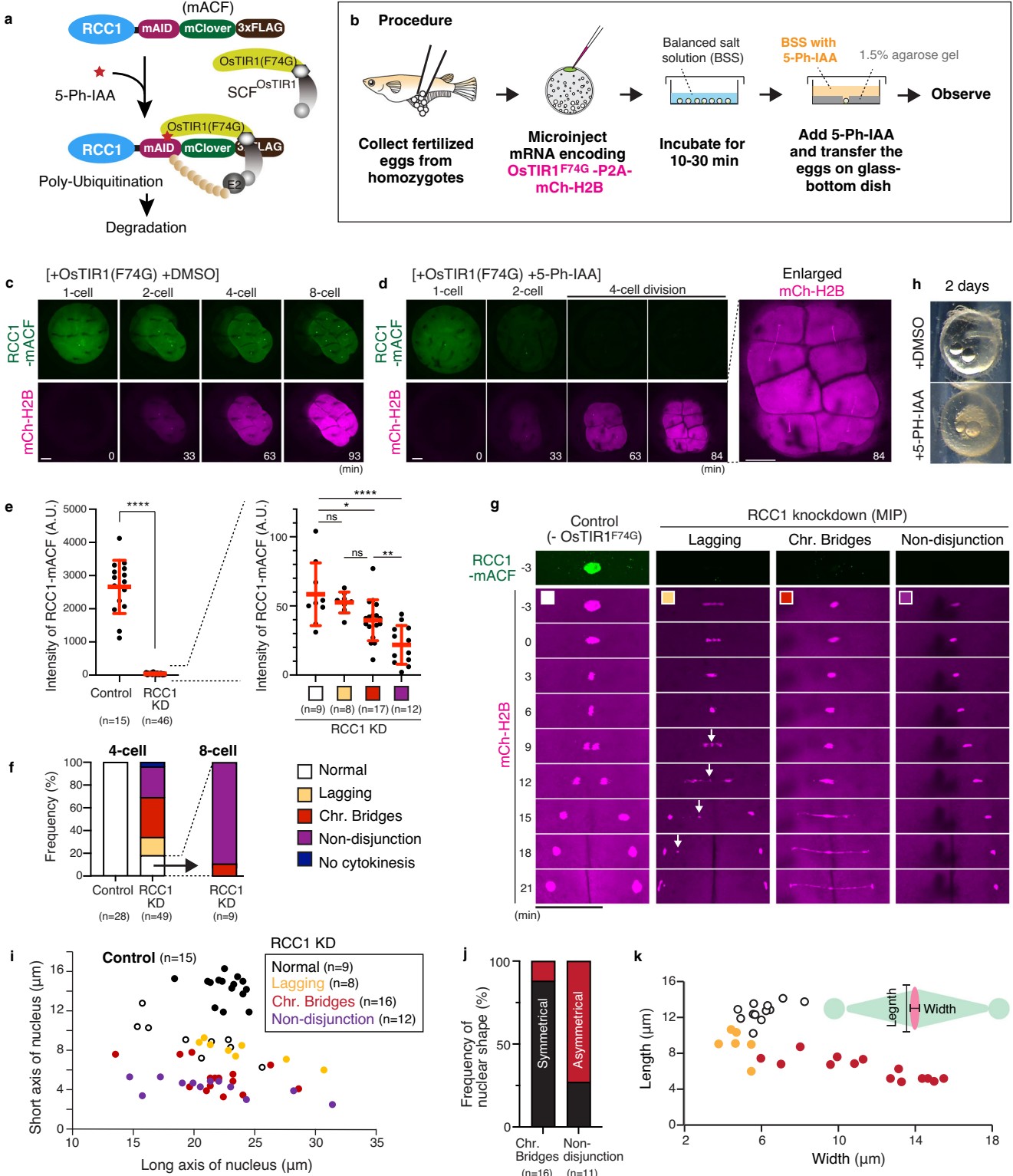

## Discussion

In this study, we characterized intracellular processes of spindle assembly, positioning, and remodeling in medaka early embryos using live imaging (Figs. 1–9), Ran mutants (Fig. 6) and AID2-mediated RCC1 degradation (Figs. 7–9). Our live imaging revealed several previously unrecognized phenomena, including comet-like structure of EGFP-α-tubulin under oil droplets (Fig. 1b), different modes of centrosome orientation (Fig. 2f), rapid and accurate spindle assembly and chromosome segregation (Fig. 4a), absence of a functional spindle

assembly checkpoint (Fig. 5h–j), centrosome separation from spindle poles without anaphase spindle elongation (Fig. 4a, b, Fig.5a–c), and transient deformed nuclear structure after mitosis (Fig. 4a) (see Supplementary Discussion).

Importantly, we found that medaka embryonic spindles assemble a specialized MT network at the spindle midplane in early stages (Figs. 3a, 3g, Fig. 10a). Before mitosis, centrosomes are located at both sides of the nucleus, and the nucleus is surrounded by MTs (Fig. 4e, Figs. 10a–1). At NEBD, unexpectedly, new MTs appear to be nucleated

**Fig. 7 | AID2-mediated degradation of endogenous RCC1 causes severe chromosome segregation defects in medaka early embryos. a** Schematic representation of auxin-inducible degron 2 (AID2)-mediated RCC1 degradation. **b** Procedure of AID-mediated protein knockdown in medaka fertilized eggs. **c**, **d** Representative live-cell images showing the fluorescence of RCC1-mAID-mClover-3xFLAG (mACF) and mCh-H2B in control (**c**) and 5-Ph-IAA-treated (**d**) embryos. **e** Quantification of fluorescence intensity of RCC1-mACF in the 4-cell-stage nuclei in control and RCC1-knockdown blastomeres (left). Right graphs show that remining RCC1 levels correlate with phenotypic severity. **f** Quantification of the frequency of abnormal chromosome segregation in control and RCC1-knockdown 4-cell blastomeres (left). Most daughter cells from normally dividing RCC1-depleted 4-cell blastomeres (*n* = 9, white) show a non-disjunction phenotype in the subsequent 8-cell division (right). **g** Live-cell images of control (left) and RCC1-depleted (3 columns on the right) 4-cell blastomeres showing chromosome

segregation phenotypes: normal (white), lagging chromosomes (yellow), chromosome bridges (red), and chromosome nod-disjunction (purple). Anaphase lagging chromosomes (arrows) result in micronucleus formation. **h** Phase contrast images 2 days after mRNA injection and treatment with DMSO (top) or 5-Ph-IAA (bottom) showing lethality in an RCC1-depleted embryo (bottom). **i** Scatter plots of nuclear size in control and RCC1-knockdown embryos. **j** Graphs showing frequency of symmetrical or asymmetrical nuclear shape at NEBD in RCC1-KD blastomeres which showed a chromosome bridge or chromosome non-disjunction phenotype. **k** Width and length of metaphase plates in **g** are plotted as colored circles for individual 4-cell spindles in control (white, *n* = 12), and metaphase cells with lagging chromosomes (yellow, *n* = 6) and chromosome bridging (red, *n* = 14). Error bars indicate mean ± SD. Scale bars = 100 μm. Source data for (**e**, **f**, **i**–**k**) are provided as a Source Data file. Two-sided Welch's t-tests were performed for (**e**). *$p < 0.1$, **$p < 0.01$, and ****$p < 0.0001$.

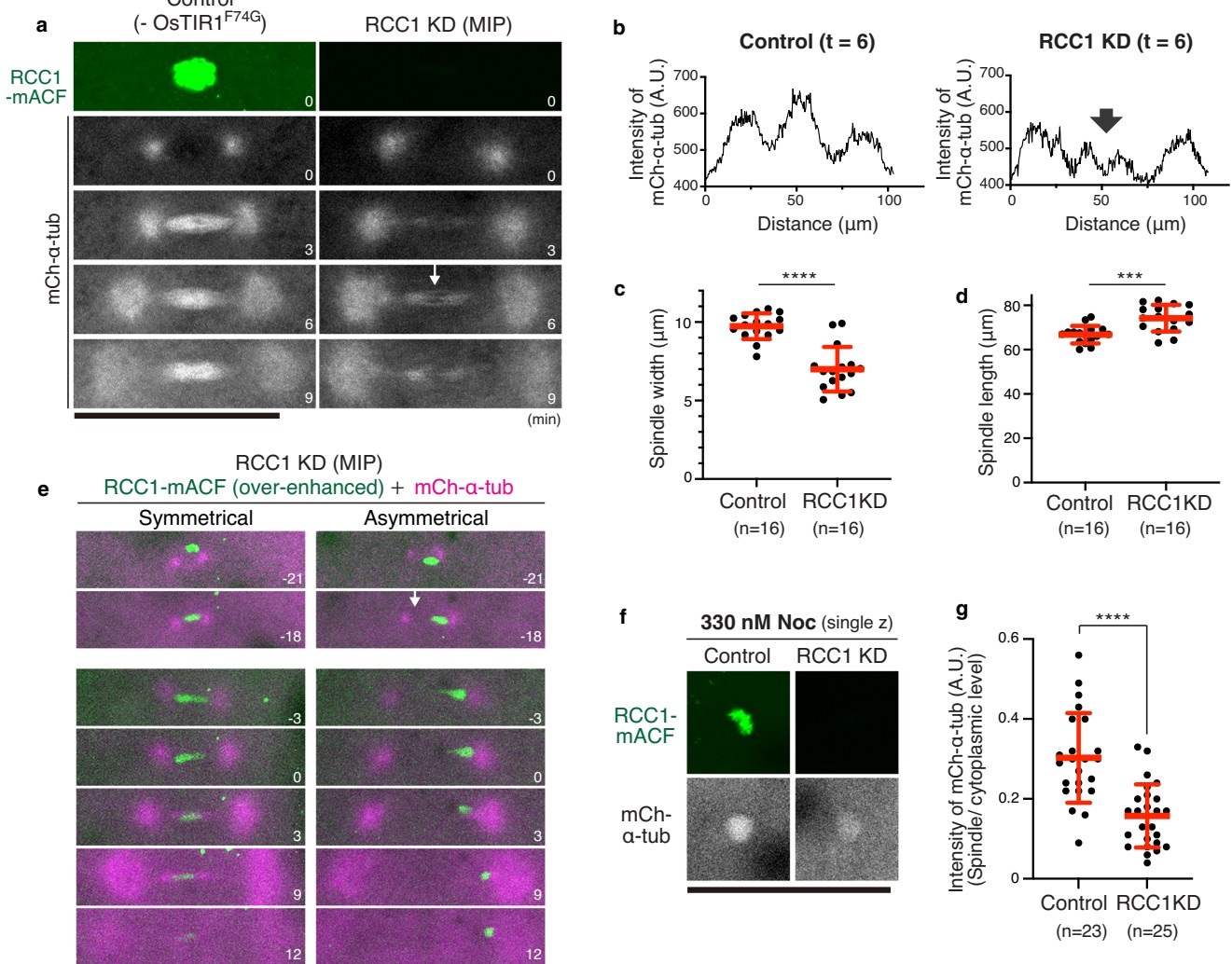

**Fig. 8 | RCC1 protein knockdown diminishes the dense MT network at the spindle center in metaphase. a** Live-cell images of control (left) and RCC1-depleted (right) 4-cell blastomeres showing disruption of the dense MT network at the spindle midplane in RCC1-depleted cells (an arrow). **b** Graphs of fluorescence intensities for line scans of mCh-α-tubulin in (**a**), showing a decrease of mCh-α-tubulin intensity at the spindle midplane in RCC1-knockdown spindles. **c**, **d** Quantification of spindle width (**c**) and length (**d**) in control and RCC1-KD

spindles. **e** Symmetrical or asymmetrical nuclear position at NEBD (t = 0) correlated with chromosome-bridge (left) and non-disjunction (right) phenotypes, respectively, in RCC1-KD blastomeres. **f**, **g** Live-cell images (**f**) and quantification (**g**) of mCherry-α-tubulin in control (left) and RCC1-KD blastomeres in the presence of 330-nM nocodazole. Error bars indicate mean ± SD. Scale bars = 100 μm. Source data for (**b**–**d**, **g**) are provided as a Source Data file. Two-sided Welch's t-tests were performed for (**c**, **d**, **g**). ***$p < 0.001$ and ****$p < 0.0001$.

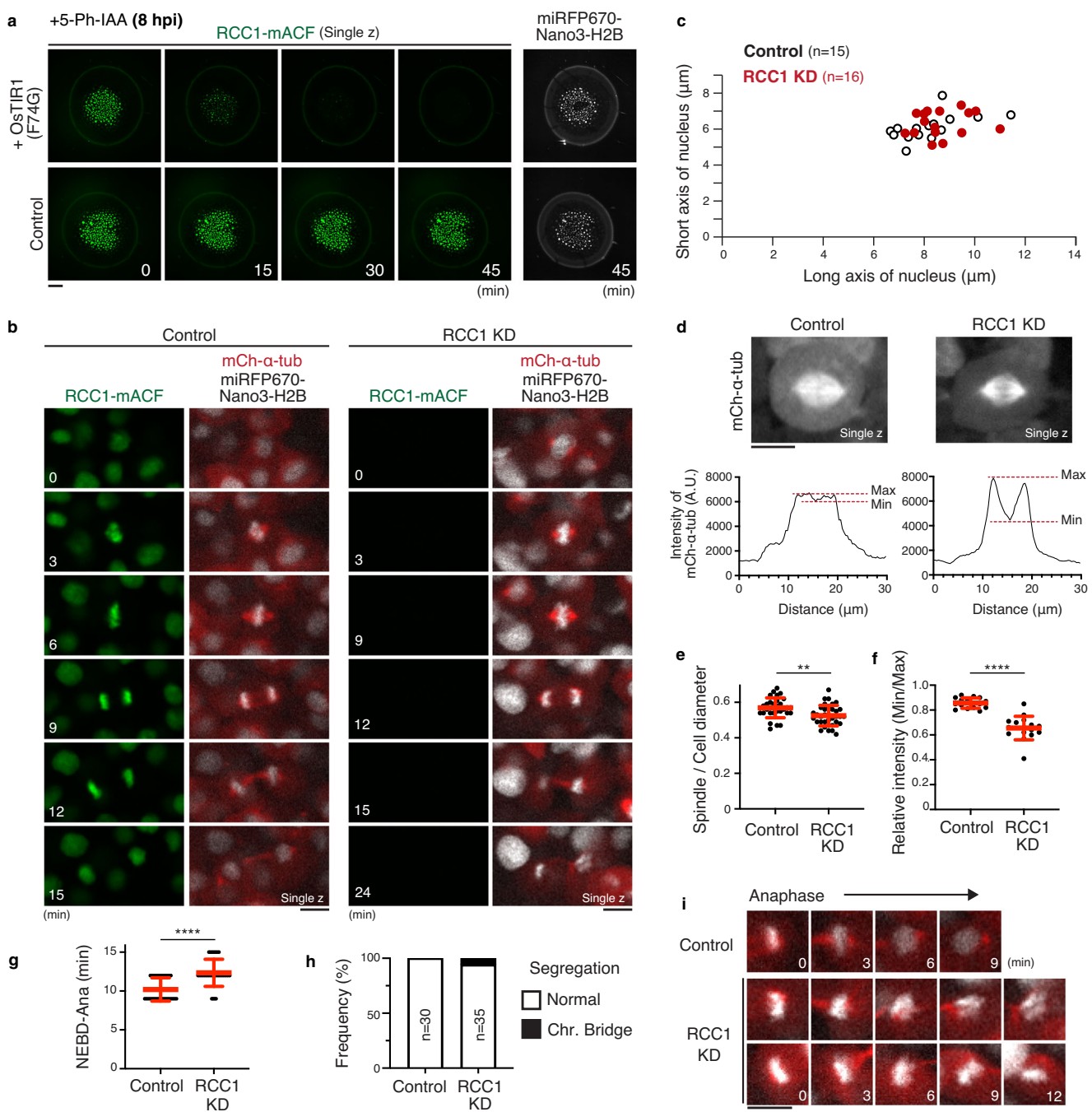

**Fig. 9 | AID2-mediated RCC1 depletion in blastula embryos causes minor defects in spindle assembly and chromosome segregation. a** Live-cell images showing fluorescence of RCC1-mACF and miRFP6700-Nano3-H2B in control (lower) and OsTIR1(F74G)-expressing (upper) embryos. 5-Ph-IAA was added 8 hr post-injection (hpi). **b** Live-cell images showing fluorescence of RCC1-mACF, mCh-α-tubulin, and miRFP6700-Nano3-H2B in control (left) and OsTIR1(F74G)-expressing (right) embryos. **c** Scatterplots showing nuclear size in control and RCC1 KD blastomeres in blastula-stage embryos. **d** Live images of mCh-α-tubulin in control and RCC1-depleted metaphase cells (top). Graphs of fluorescence intensities for line scans of mCh-α-tubulin in (**d**), showing a decrease of mCh-α-tubulin intensity at the spindle midplane in RCC1-knockdown spindles. **e** Ratio of spindle length and cell diameter in control (*n* = 30) and RCC1 knockdown (*n* = 35) cells. **f** Relative intensity (Min/Max) of mCh-α-tubulin fluorescence on the spindle in control (*n* = 15) and RCC1-KD blastomeres (*n* = 16). **g** Scatterplots of mitotic duration in control (10.2 ± 1.5, *n* = 30) and RCC1-depleted (12.3 ± 1.7, *n* = 35) cells. **h** Quantification of normal and abnormal anaphase with chromosome bridges in control and RCC1-depleted cells. **i** Representative live-cell images showing nuclear reformation defects in RCC1 knockdown cells. Error bars indicate mean ± SD. Scale bars = 100 μm (**a**) and 10 μm (**b, d, i**). Source data for (**c–h**) are provided as a Source Data file. Two-sided Welch's t-tests were performed for (**e–g**). **\**p* < 0.01 and \*\*\*\**p* < 0.0001.

around chromosomes (Fig. 4e-h, Figs. 10a-2), and these MTs around chromosomes and the nuclear membrane seem to trap chromosomes inside the cage-like structure (Figs. 10a-2). Subsequently, these MTs are clustered near centrosomes, likely with parts of astral MTs, to form focused spindle poles (Fig. 4a t = 1-2, Figs. 10a-3). During prometaphase, new MTs seem to be generated at a whole spindle region (Fig. 4a t = 4, Figs. 10a-4). These MTs seem to be bundled and stabilized preferentially around chromosomes in concert with chromosome biorientation, and then finally organize a dense MT network at the spindle center in metaphase (Fig. 4a t = 5–8, Figs. 10a-5). This

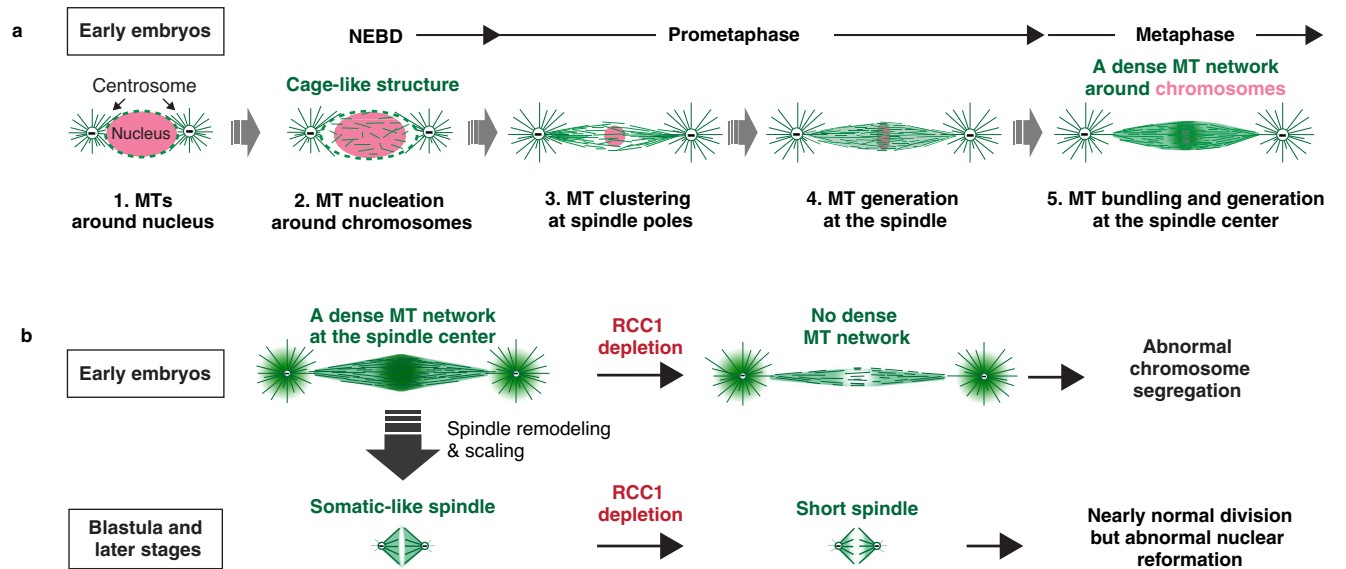

**Fig. 10 | Spindle assembly in medaka early embryonic divisions. a** MT organization and spindle assembly processes from prophase to metaphase in medaka early embryos. **b** Diagram showing remodeling of spindle architecture and the requirement for RCC1 in early and later-stage embryos. See text for details.

central MT network can be formed in the absence of robust astral MTs (Fig. 4h), and consists of multiple MTs, including bundled k-fibers (Fig. 3i), bridging fibers (Fig. 3g, i), dynamic EB1-positive MTs (Fig. 4f), and unstable MTs sensitive to cold treatment and our fixation protocol (Fig.3 f–h). Since centrosomes are largely separated from the spindle midplane (>30 μm) after metaphase (Fig. 4a, Fig. 5a, b), chromosome-derived Ran-GTP signals apparently act as an essential factor to assemble functional spindles for chromosome segregation in early embryos, like acentrosomal spindle assembly in oocytes[31–33]. A dense midplane MT network is observed in zebrafish early embryonic spindles[7], and *Xenopus laevis* stage 3 spindle intermediates[3], but not clearly in bovine 1- or 2-cell spindles[16,60]. In *Xenopus* egg extracts, TPX2 levels modulate MT density at the spindle center and poles via Eg5[61]. TPX2 loading on spindle MTs also influences spindle architecture in mouse neural stem cells[62]. In future investigations, it is critical to understand how TPX2 level is controlled[63] and how Ran-GTP regulates activities of TPX2[28,64,65] and other Ran-regulated SAFs including HURP[34,66], KIFC1[67,68] and augmin[69–71] to control MT nucleation (Fig. 4e–h), MT bundling and organization for embryonic spindle assemblies in medaka (Fig. 10a) and other organisms. In addition, since medaka embryonic divisions rapidly cycle without checkpoint controls (Fig. 5h, i), accuracy of earlier events such as nucleus-centrosome interaction appear to become more important to achieve accurate spindle assembly and positioning (Fig. 2d, e, Fig. 8e).

On the other hand, we also found that Ran's contributions to spindle assembly change dramatically in response to remodeling of spindle structure during embryogenesis in medaka (Fig. 10b), consistent with a previous study with *Xenopus* egg extract[3]. As observed in a somatic cultured human cell line[34], Ran-GTP becomes non-essential for spindle assembly and chromosome segregation in later-stage embryos (Fig. 9), in which Ran is essential for nuclear reformation after mitotic exit[72]. Other Ran-independent chromatin pathways such as the chromosome passenger complex[73–76] and centrosomes should be sufficient to form functional bipolar spindles in smaller cells after blastula stage (Fig. 9b, Fig. 10b).

In summary, this study provides a foundation to understand mechanisms of chromosome segregation and spindle assembly, positioning, and remodeling in medaka early embryos. Studying their detailed mechanisms is important to identify general and species-specific strategies that successfully coordinate chromosome segregation with cell differentiation and embryogenesis.

## Methods

### Fish maintenance
Fish experiments were conducted in accordance with protocols (2019-273-5, 2022-369) approved by the Animal Care and Use Committee at Okinawa Institute of Science and Technology Graduate University (OIST). The OK-Cab strain (MT830) of medaka (*Oryzias latipes*) was obtained from the National Bio-Resource Project Medaka (NBRP Medaka) and used as the parental strain. Medaka were maintained in fresh water at 26–28°C under a regulated photoperiod (14-h light and 10-h dark). Medaka embryos were cultured at 27-28°C. Medaka larvae were grown in a tank without water circulation for 2-3 weeks and then transferred into a tank with water circulation (Meito system, Meito Suien or ZebTEC, Tecniplast). Fish were fed three times per day with live brine shrimp at 1 pm, and a commercial dry feed (Hikari-Lab, Kyorin) around 9 am and 5 pm. For larvae, paramecia (PS000001A), provided by NBRP Paramecium Laboratory, were provided with dry feed for the first 3–5 days. From 4–6 days, live brine shrimp were fed instead of paramecia. The particle size of dry feed (Hikari-Lab, 130,270,450) was adjusted according to fish size.

Naturally fertilized eggs were collected from 2–9 month old pairs. Healthy fertilized eggs were used for imaging. Fluorescence of RCC1-mCh and RCC1-mACF was constantly observed in all heterozygous and homozygous knock-in embryos. However, fluorescence of EGFP-α-tubulin was variable, even in homozygous embryos and was sometimes undetectable for unknown reasons. We selectively used embryos having similar fluorescence intensity of EGFP-α-tubulin in all experiments. Established plasmids, Medaka strains, and sequence information about guide RNA (gRNA) and PCR primers used in this study are described in Supplementary Table 1, 2, 3 and 4, respectively.

### Plasmid Construction
Donor dsDNAs for CRISPR/Cas9-mediated genome editing (Supplementary Fig. 1a and 7a) were constructed according to the protocol of Gutierrez-Triana et al.[45]. To design a donor plasmid, PAM sequences were searched around the stop codon of the medaka (*Oryzias latipes*) RCC1 gene (Ol-RCC1) on chromosome 16 using CCTop, CRISPR/Cas9 target online predictor (https://cctop.cos.uni-heidelberg.de:8043/)[77]. Silent mutations were introduced to the gRNA target and PAM sequences. The stop codon was mutated to encode glycine (G), and a BamHI site was introduced in-frame after the G so that CDS cassettes

flanked by BamHI could be inserted. The designed DNA containing these mutations, a BamHI site and homology arms for Ol-RCC1 (~450-bp homology arms) was synthesized by a gene synthesis service (Genewiz, South Plainsfield, NJ). mCherry2 and mAID-mClover-3xFLAG(mACF) cassettes were created using pMK281 (addgene #72797) and pTK398 (addgene #114714), respectively, and inserted at the BamHI site on the synthesized DNA plasmid to make donor plasmids, pTK997 and pTK1023. Using these plasmids as a template, dsDNAs were amplified by PCR using modified primers with 5'Bioton – 5 x phosphorothioate bonds (synthesized by eurofins) and PrimeSTAR Max (Takara).

To construct a knock-in (KI) donor vector for EGFP-α-tubulin (Supplementary Fig. 1b), Mbait-hsp-EGFP-BGHpA plasmids[78] were linearized by PCR using KOD-Plus-Neo (TOYOBO), followed by insertion of a flexible linker[79] at the C-terminus of EGFP using the InFusion system (Takara). α-tubulin CDS was amplified from a cDNA clone, olli34h17, (gifted by NBRP Medaka), and inserted into the XbaI site of resultant Mbait-hsp-EGFP-FL-BGHpA plasmids to generate Mbait-hsp-EGFP-FL-α-tubulin-BGHpA (KI vector). A CRISPR/Cas9 target site at 5'UTR of α-tubulin was searched using CCTop.

To exogenously express EGFP- or mCherry-fusion proteins, CDS of EGFP-EB1 (addgene #46364) and EMTB-3xGFP (addgene #26741) were inserted between BamHI-SnaBI sites in pCS2+hSpCas9 vector (addgene #51815, a gift from Dr. Kinoshita at Kyoto University). H2B CDS was obtained from pSNAPf-H2B (New England Biolabs). CDS of Ol-RanT27N and miRFP670-Nano3 was synthesized by eurofin, and Ol-RanWT CDS was created by PCR-based mutagenesis.

To make OsTIR1-P2A-mCh-H2B and OsTIR1-P2A-mCh-α-tubulin plasmids, pMK411 (addgene #140659) was used as a template to amplify the CDS of OsTIR1-P2A, which was fused with mCh-H2B or mCh-α-tubulin by PCR. CDSs were inserted between BamHI-SnaBI sites of the pCS2+hSpCas9 vector.

### gRNA synthesis

For synthesis of gRNA, T7 tagged DNA templates containing the gRNA sequence were amplified by PCR, as described previously[80]. For synthesis of Mbait gRNA, a DR274 vector (addgene # 42250) containing Mbait (gifted by Dr. Yasuhiro Yamamoto at Osaka Medical and Pharmaceutical University) was used as a template to amplify fragments containing T7 and gRNA sequences by PCR followed by gel purification. PCR fragments were used as a template for in vitro transcription with a MEGAscript T7 kit (Thermo Fisher Scientific, AM1333) according to the manufacturer's instructions. Synthesized gRNAs were purified by ammonium acetate precipitation.

### In vitro transcription of mRNA

For synthesis of Cas9 mRNA, the pCS2+hSpCas9 vector was linearized by NotI digestion, followed by in vitro transcription using the mMessage mMachine SP6 kit (Thermo Fisher Scientific, AM1340) according to the manufacturer's instructions. The synthesized RNA was purified with an RNeasy Mini kit (Qiagen). To synthesize other mRNAs, template plasmids were linearized with NotI or BssHII.

### Microinjection

To generate RCC1-mCh (Supplementary Fig. 1a) or RCC1-mACF (Supplementary Fig. 7a) knock-in strains, 50 ng/μL gRNA targeting RCC1, 150 ng/μL Cas9 mRNA and 10 ng/μL dsDNA were injected into one-cell stage medaka embryos. To establish an EGFP-α-tubulin knock-in strain (Supplementary Fig. 1b), 25 ng/μL Mbait gRNA, 25 ng/μL gRNA targeting α-tubulin, 50 ng/μL Cas9 mRNA and 2.5 ng/μL KI vector were used. To exogenously express EGFP- or mCherry-fusion proteins, 150 ng/μL mRNAs were injected into one-cell embryos.

Glass needles were made from borosilicate glass capillaries (Model No: G100F-4, Order No: 64-0787, Warner Instruments) using a needle puller (PC-100, Narishige), and attached to a capillary holder connected with a microinjector Femto Jet 4i (eppendorf). Needles were manually controlled by a micro-manipulator (MN-153, Narishige) on the stage of a stereomicroscope (Leica M80).

### Genotyping and selection of knock-in strains

Small pieces of tail fin were cut and added to 20 μL lysis buffer (0.1 M Tris-HCl, 0.2 M NaCl, 0.1% SDS, 5 mM EDTA). After 5 min incubation at 95°C, 80 μL $H_2O$ were added. After centrifugation, 1-μL aliquots of supernatant were used for genomic PCR with KOD FX Neo (TOYOBO) and appropriate primers (Table 4). Candidate G0 fish were crossed with wild-type counterparts, and fertilized eggs were analyzed for fluorescence of RCC1-mCh or RCC1-mACF, 1 or 2 days after fertilization. Genomic DNA was extracted from some fluorescence-positive F1 embryos, and genomic PCR was performed. Proper knock-in of the construct was confirmed by direct sequencing of PCR products. Other positive embryos from the same G0 fish were reared, and F1 male and female fish were mated to obtain a homozygous F2 generation. PCR products were analyzed by normal gel electrophoresis or using an automatic microchip electrophoresis system (MCE-202 MultiNA: Shimazu, Kyoto, Japan).

To select the EGFP-α-tubulin knock-in strain (Supplementary Fig. 1b), embryos with EGFP fluorescence in their eyes and throughout their bodies were raised as G0 founders and were crossed with wild-type medaka to obtain a stable F1 transgenic line. Insertion of the KI vector at the target site was examined by PCR using specific primers for the KI vector and the promoter of α-tubulin (upstream of gRNA target site).

### Microscope system and live imaging

Imaging was performed with spinning-disc confocal microscopy with a 20× 0.95 numerical aperture objective lens (APO LWD 20X WI λS, Nikon, Tokyo, Japan). A CSU-W1 confocal unit (Yokogawa Electric Corporation, Tokyo, Japan) with three lasers (488, 561, 640 nm, Coherent, Santa Clara, CA) and an ORCA-Fusion digital CMOS camera (Hamamatsu Photonics, Hamamatsu City, Japan) were attached to an ECLIPSE Ti2-E inverted microscope (Nikon) with a perfect focus and a water-supply system. Images were captured using NIS-Elements software (version 5.21.00, Nikon). Samples were imaged at room temperature (24-25°C).

To hold living embryos for time-lapse imaging, a handmade device was created: 1.5% agarose solution was added to glass-bottomed dishes (CELLview™, #627860, Greiner Bio-One, Kremsmünster, Austria), and a 7-unified cover-glass (18 mm×18 mm with 0.12–0.17 mm in thickness, Matsunami) was set as a mold in the center of the glass-bottomed dish. When the agarose became solid, the unified glass was removed to make a concave pocket in the agarose gel, 0.8–1.2 mm in width and 3–5 mm in height on the bottom glass (Fig. 7b). 2–8 embryos were put in the pocket in a row with the blastodisc facing the glass (Fig. 1a). The dish was filled with ~2 mL medaka balanced salt solution (BSS, 0.65% NaCl, 0.04% KCl, 0.02% $MgSO_4 \cdot 7H_2O$, 0.02% $CaCl_2 \cdot 2H_2O$, sterilized and adjusted to pH 8.3 with 5% $NaHCO_3$) without phenol red. 11–13 z-section images 5 μm thick were acquired every 3 min or 1 min with camera binning 1. Green and red fluorescence and DIC images were captured in this order with exposure times of 500 msec, 1 sec, 100 msec, respectively. X-Y-Z positions of 2–7 embryos were memorized and automatically imaged for 6–10 h in a time-lapse experiment. For figures, 8-bit maximally projected z-stack images (MIP) or single z-section images are shown as indicated. MIP images were created using NIS-Elements or Fiji. Signals were linearly adjusted using Fiji and Photoshop to optimize image clarity, and images were arranged using Adobe Illustrator. Phase-contrast images of embryos for Fig. 7h and Supplementary Fig. 7d were taken with a Leica M80 stereo microscope and a Leica MC190 HD camera.

## AID2-mediated protein knockdown

For AID2-mediated protein degradation, mRNAs encoding OsTIR1(F74G)-P2A-mCherry-H2B or OsTIR1(F74G)-P2A-mCherry-α-tubulin were injected into one-cell embryos. Injected embryos were cultured in BSS without phenol red for 10–30 min with gentle shaking. The BSS was replaced with 2 mL BSS containing 10 μM 5-Ph-IAA or 0.1% DMSO, and then these embryos were transferred with the solution to the handmade glass-bottom dish. After adjusting the orientation of eggs in the agarose pocket, eggs were observed by microscope (Fig. 7b). For experiments in Fig. 9, injected embryos were cultured in medaka BSS for 8 hr before 5-Ph-IAA treatment.

## Nocodazole treatment of medaka embryos

Collected fertilized eggs were mechanically separated from attachment filaments using fine forceps (DUMONT, No.5-INOX), and incubated in a 24-well plate containing 170-nM or 330-nM nocodazole (Sigma-Aldrich, M1404) in BSS for 10–30 min at 27.5°C. Then, eggs were moved into a hand-made agarose pocket on the glass-bottom dish containing BSS with nocodazole. To combine the nocodazole treatment with AID-mediated RCC1 degradation in Fig. 8f, g, embryos were treated with 330 nM nocodazole for 30 min before transferring them into glass-bottom dishes containing both 5-Ph-IAA and nocodazole.

## Fixation and immunofluorescence of medaka embryos

Embryos were selected at an appropriate stage using a spinning-disc confocal microscope and fixed overnight with 4% paraformaldehyde (FUJIFILM Wako, 168-23255) in PBS + 0.1% Tween20 (PTw) at room temperature. Fixed embryos were rinsed three times with PTw then the chorion was removed mechanically with fine forceps and a surgical blade (Feather No.11). Dechorionated, fixed-embryos were washed three times with PTw for 10 min each, followed by acetone treatment at −20°C for 20 min. Embryos were washed with PTw 3 ×5 min and then incubated in blocking buffer (3% Bovine Serum Albumin (Sigma-Aldrich, A7030), 0.8% Triton X-100 in PTw) for 1 h at room temperature. After blocking, embryos were treated with anti-α-tubulin mouse monoclonal antibody (Sigma-Aldrich, T6199) or anti-γ-tubulin mouse monoclonal antibody (Sigma-Aldrich, T6557) at a 1:200 dilution in blocking buffer at 4°C overnight (>12 h) with gentle shaking (TAITEC, invitro shaker Shaker XR). After washing with 0.8% Triton X-100 in PTw for 3 ×20 min at room temperature with gentle shaking, samples were incubated with secondary antibodies goat anti-mouse IgG Alexa Fluor Plus 555 (Invitrogen, A32727) or goat anti-mouse IgG Alexa Fluor Plus 647 (Invitrogen, A32728) and Hoechst 33342 (FUJIFILM Wako 346-07951) in the dark at 4°C overnight (>12 h) with gentle shaking. All secondary antibodies and Hoechst 33342 (100 μg/mL) were incubated at 1:1000 dilutions in blocking buffer. After washing 3 ×20 min in 0.8% Triton X-100 in PTw in the dark at RT with gentle shaking, embryos were put in the agarose pocket (using a 6-unified cover-glass as a mold) on a glass-bottomed dish filled with 0.8% Triton X-100 in PTw. Finally, embryos were observed under spinning-disc microscopy.

## Cold treatment and MT re-growth assay

Embryos at an appropriate stage were selected using the spinning-disc confocal microscope and moved from the agarose-channel in glass-bottomed dishes to on-ice dishes with pre-chilled BSS. After 20–30 min, one or two embryos were quickly replaced in the pre-chilled agarose-channel in glass-bottomed dish filled with pre-chilled BSS and observed using the spinning-disc microscope at room temperature. 5–7 z-sections with 5 μm spacing were acquired every 15–20 sec with camera binning 1. Green and red fluorescence images on the same z-sections were sequentially captured in this order using exposure times of 500 msec and 1 sec, respectively. 2-cell embryos having metaphase spindles were selected for Fig. 3h, whereas embryos that had just entered mitosis were chosen for Fig. 4i to completely destabilize spindle MTs around chromosomes for the MT regrowth assay. In the MT regrowth assay, only one embryo was imaged per experiment for 20 min.

## Quantification of length, duration, fluorescent intensities and phenotypes

For line scans and measurement of metaphase spindles, metaphase spindles having both centrosomes/poles in a single-focal plane were selected (Figs. 3b–f, 4c, 5c, 9d–f and Supplementary Fig. 3b). For Fig. 3f and Supplementary 3c, the max values of EGFP-α-tubulin intensity on spindle MTs and astral MTs were determined by measuring several z-sections. MIP images were used for Figs. 6c and 8b. To quantify fluorescence or immunofluorescence intensities of EGFP-α-tubulin, EGFP-EB1, EMTB-3xGFP, mCh-α-tubulin, and anti-α-tubulin, line scans were performed in Fiji (version 2.3.0/1.53q) or NIS-Elements (version 5.20.00, Nikon). A line 5-pixels wide was drawn as it passed through both spindle poles and centrosomes (Fig. 3b). A line 3-pixels wide was used for Fig. 9d and f. NIS-Elements software (version 5.20.00 or 5.40.00) was used to measure mitotic and cell-cycle duration (Fig. 2b, c), cell sizes (Supplementary Fig. 2b) or nuclei (Figs. 6d, 7i, 9c, Supplementary Fig. 2b), centrosome movement (Fig. 2g), and spindle length and width (Fig. 3c, d). For Figs. 5b, 5l, positions of chromosomes, spindle poles, and centrosomes were defined based on the intensity profile of line scans on MIP images as indicated in Figs. 3b and 5c. Mean fluorescent intensities of RCC1-mACF in the nucleus one frame before NEBD were measured using NIS-Elements. mCh-H2B images were used as a reference when measured regions were determined. Mean fluorescent intensities of mCh-α-tubulin were measured by manually determining the spindle region in Fig. 8g. Neighboring cytoplasmic fluorescence intensities were subtracted in Figs. 7e and 8g. Graphs were created using Prism 10 (version 10.0.2, GraphPad Software, La Jolla, CA) or Excel (version 16.61.1, Microsoft).

## Illustration

Diagrams shown in Figs. 1a, 2c, 2f, 4b, 5m, 6a, 7a, b, 10 and Supplementary Fig. 1a, b, and 7a were created using adobe illustrator 2022 (version 26.3.1, Adobe).

## Statistics and Reproducibility

GraphPad Prism version 10 (version 10.0.2) was used. Mean, SD, and statistical significance were determined using two-sided Welch's t-tests for Fig. 5h, Fig. 7e, Figs. 8c, d, 8g, Fig. 9e–g, and Supplementary Fig. 3b. One-way ANOVA with Dunnett's multiple comparisons test was performed for Fig. 2b, Figs. 3c, d, 3f, Fig. 7e, and Supplementary Fig. 3a. P values are shown as *: *: $p < 0.1$, **: $p < 0.01$, ***: $p < 0.001$ and ****: $p < 0.0001$. At least 3 independent experiments were performed for Figs. 1b–e, 2d, 3g, 3i, 4e–i, 5d–f, 6b, 6e, f and 8e, and similar results were obtained in the repeated experiments. No statistical method was used to predetermine sample size. Sample sizes were chosen based on standards in the field and are sufficient to do appropriate statistical tests. For quantification of cell cycle and spindles in Figs. 2b, c, 2g, 3b–e, 4c-d, 5b, c, and Supplementary Fig. 2b, c, and 3a-c, embryos that had hatched normally after imaging were selectively used to understand physiological intracellular dynamics. The healthy fertilized eggs were used randomly. The investigators were not blinded for data collection and quantification.

## Reporting summary

Further information on research design is available in the Nature Portfolio Reporting Summary linked to this article.

# Data availability

All data supporting findings of this study are available in the paper and its Supplementary Information. Source data for each graph are

provided with this paper. All data of this study are stored at the corresponding author and available on reasonable request.

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

## Acknowledgements

We thank Kiyoshi Naruse, Ayaka Mori, Yoko Nakasone and the OIST animal resource section for technical assistance and support, and Marvin van Toorn for critical reading of the manuscript. We are grateful to NBRP Medaka (https://shigen.nig.ac.jp/medaka/) and NBRP Paramecium Laboratory (http://nbrpcms.nig.ac.jp/paramecium/) for providing OK-Cab (Strain ID: MT830) and *Paramecium* strain (PS000001A), respectively. This work was supported by grants from JSPS KAKENHI (17H05002 and 21H02481 to TK, and JP21H0419 and JP23H04925 to MTK), JST FOREST (JPMJFR224O to TK) and CREST (JPMJCR21E6 to MTK), the Takeda Science Foundation (to TK), the Uehara Memorial Foundation (to TK), and the Okinawa Institute of Science and Technology Graduate University, Japan.

## Author contributions

Conceptualization: TK. Investigation: AK and TK. Formal analysis: AK and TK. Methodology: AK, TN, SA, MTK, MT, and TK. Rearing and maintenance of Medaka fish: AK, SH, and TK. Writing: TK. Supervision: TK. Funding Acquisition: TK. and MTK.

## Competing interests

The authors declare no competing interests.
