## [Peer Review File · Nature Communications]

REVIEWER COMMENTS

Reviewer #1 (Remarks to the Author):

Kiyomitsu and colleagues hand in a manuscript describing the analysis of RanGTPs' role in the assembly of "a specialized spindle structure for accurate chromosome segregation".

The authors exploit the established model system of medaka fish and use technically challenging live embryo imaging with chromatin and tubulin markers to revisit the role of RanGTP in embryonic spindle formation and chromosome segregation. They take on overexpression of Ran (nucleotide binding def.) and RCC1-AID gene replacement for RCC1 protein knockdown. In the first part of the manuscript, they show a thorough description of spindle formation and chromosome segregation in synchronous medaka embryo blastocyst. This reveals not only two topologically different spindle assembly pathways but demonstrates the existence of a super-dense MT structure in the central part of early spindles. They further show an early window of strong Ran-dependence in the first divisions while later during development the system requires less/no RanGTP production for spindle formation indicating that the initial dense MT core in early spindles is generated by, or at least requires, RanGTP activity.

To say it from the very beginning: I am very enthusiastic about this work for three main reasons: First, it is technically excellent, all experiments are down extremely carefully and with tailor-made tools to address the hypotheses generated. Second, the initial observation of the dense MT core in early spindle of medaka embryos is conceptually interpreted exactly right, i.e. associated to RanGTP activity. And, third, the work provides a reasonable explanation as to why previous data on Ran's role in spindle formation as focused on embryonic situations while being less hard to grasp in somatic cells.

Having said this, I am truly in favor of publishing the paper in its present form, possibly after some minor issues concerning the writing of the manuscript (see below) have been addressed.

Abstract:

The central statement quoting a "gradient" is to my opinion an overstatement. We all believe that RanGTP distributes in a gradient-like fashion but this is not addressed here "activity of" "high concentration of..." or just "RanGTP" (see end of intro) would do instead. I would suggest to carefully check the wording in the manuscript throughout accordingly.

p.8: "synchronous" rather than "synchronized"

sentence lanes 182/3 is not completely logical as it compares timing "down" to more cells and timing "up" to fewer cells when saying "to reach an upper limit..."

lanes 206/7 and 272 MT assembly instead of nucleation

lane 253: corresponds to mouse and human RanT24N

lane 380: clearly

Articles „the“ / „a“ if Fig. 1A; Microtubules;

Fig. 1B: insets may be indicated in the larger views (see 1D);

2G: small n.

3D: graph: I suggest going down to 0 in the ordinate to highlight the relatively small changes in width as compared to size

Fig. 7 D and E: I am not a friend a having non-0 ordinates...

7H: possibly highlighting the region which is most dependent of RanGTP/RCC1 activity levels ("dense MT network...") in the central part of the spindle would be an advantage

Reviewer #2 (Remarks to the Author):

In this study, the authors convincingly describe in live mitotic spindle morphogenesis of the early medaka embryo and its evolution over time. They show that mitotic divisions are very rapid but faithful, in contrast to mammalian embryos. They suggest with some functional tests that a Ran-GTP gradient assembles a dense microtubule network around metaphase chromosomes, essential for correct chromosome segregation. Interestingly, Ran-GTP becomes dispensable at later embryonic stages, correlating with the change in spindle morphology, spindles becoming shorter, with few astral microtubules, and no dense microtubule network around metaphase chromosomes. I agree with the authors that their study provides the foundation to understand mechanisms of chromosome segregation and spindle assembly, positioning, and remodeling in medaka early embryos. However, if this paper provides a nice description of the events, some statements are not supported by the data and thus too strong, lacking a deeper analysis, quantifications and statistical tests. In addition, the functional tests only skim the surface and should go deeper into the mechanisms at work. Please find below my comments, which are intended to improve the quality of the manuscript.

Descriptive part of the paper (Figures 1 to 4):

- It would be nice to show a centrosome staining instead of inferring where they are by the tubulin signal, especially for describing the Type I and II configurations of centrosome orientation (Figure 2F), and for Figure 3A-B.
- The centrosomes move apart from each other in metaphase (Figure 4A). For example, centrosomes in 1-cell stage move apart between 027 and 033 min (same for 2-cell stage between 063 and 069 min, and for 4-cell stage between 096 and 099 min, see Movie S2). This is less the case later on. Typically, in Figure 3A-B, when are the linescans positioned? It could matter since spindle length is defined here as the centrosome-centrosome distance. In addition, few spindles were observed at that stage: N=5 for 1-cell stage, N=9 for 2-cell stage, N=15 for 4-cell stage. These numbers should be increased, and analyzed with appropriate statistical tests.
- The authors say that spindle width and metaphase plate length are relatively constant until the early blastula stage (Figure 3D, S3A). This should be addresses using appropriate statistical tests.
- The authors conclude that since EGFP-EB1 is localized at the spindle center (Figure 3F), it means that a specialized MT network is formed around chromosomes in early spindles. Could the authors do cold treatments and then regrow of MTs, to show that they are indeed nucleated from the chromosomes? And if this network is nucleated by Ran-GTP, what is their hypothesis regarding the fact that MTs are not

nucleated earlier on the chromosomes, after NEBD? The cold assays could also test the nucleation capacity of the centrosomes, that could decrease with divisions and explain the disappearance of astral microtubules with time.

- EMTB-3xGFP is used as a control, because it does not accumulate at the spindle center. Looking closely at the images (Figure 3 F-G), one could wonder if the cells are exactly at the same stage. Indeed, the metaphase plate is thinner in Figure 3G compared to Figure 3F. Could it be that the spindle in Figure 3F already initiated anaphase, and maybe started nucleating microtubules between the DNA (as it is the case in nematode oocytes), whereas the spindle in Figure 3G is in metaphase? At last, the observation that EMTB-3xGFP recognizes signals around cortical cell-membranes does not mean that medaka early embryonic spindles have unique structures that consist of MTs with different characters.

- It could be worth redoing some of the key experiments in Immunofluorescence to make sure that expression of tagged proteins does not modify their localization.

- Overall, quantifications are done on a low number of embryos, usually not more than 4 or 5. It should be increased, as it is difficult to do statistical tests on such small numbers.

- A lot of interesting observations are reported but not quantified nor commented, such as the Type I and II configurations of centrosome orientation, the bended spindles (could it just be that the spindle is a little bit tilted in respect with the polarity axis of the cell?), comet-like structure under oil droplets, difference in cell cycle length between inner and peripheral blastomeres, MT-like signals around cortical cell membranes, lack of anaphase elongation, centrosome-centrosome increase in distance in metaphase, transient deformed structures, absence of spindle assembly checkpoint, polyploid-like nuclei, spindle scaling with cell size after the 16-cell stage and upper limit in the fourth divisions...

Functional part of the paper (Figures 5 to 7):

- The authors are interested in Ran-GTP, but they never show that there is a Ran-GTP gradient. The authors should use FRET probes (cf Kalab 2006) to demonstrate that there is a Ran-GTP gradient, and how this gradient evolves during time along with the switch in spindle morphogenesis observed by the authors. Then, this gradient should be measured after OI-RanT27N injection and RCC1 depletion via auxin degra.

- The authors manipulate Ran-GTP levels, using OI-RanT27N, that inhibits the generation of Ran-GTP. Why OI-RanT27N would cause deformation or elongation of nuclei? Could they do the reverse, using the equivalent of the Ran Q69L, which stays locked in the GTP-bound state?

- I don't understand the rationale for doing the aphidicolin experiment, I would remove it.

- Quantifications are missing for lots of experiments, so it is hard to grasp the main phenotype following OI-RanT27N injection. How much cells were analyzed in total? How many displayed precocious detachments of a centrosome (some means a lot?), and why would a centrosome detach from the nucleus without Ran-GTP? How many displayed asymmetric spindles? How many had chromosome segregation defects, and what kind of defects (as in Figure 6G)? Does it cause embryonic lethality as for RCC1 depletion?

- The authors conclude that chromosomes are required for spindle assembly, but are 2 centrosomes without chromosomes always making a spindle? I don't understand their conclusion, that Ran-independent chromatin pathways collaborate with centrosomes in spindle MT nucleation near centrosomes. Ran-GTP pathway can regulate microtubule assembly/dynamics, so without it, it is not surprising that less MTs are produced.

- Why OI-RanT27N expression and RCC1 depletion do not cause the same defects, with stronger defects for RCC1 depletion (such as chromosome non-disjunction)? Could RCC1 depletion via auxin degraon impair chromosomes, and not only the spindle, leading to some defects in chromosome segregation, since RCC1 is localized on the chromosomes?

Minor comments:

- In the introduction, the authors say that the Ran-GTP gradient is essential for acentrosomal spindle assembly in female meiosis. They should be more cautious, since one paper shows in mouse and xenopus oocytes that the Ran-GTP gradient is dispensable in Meiosis I (spindles can form and bipolarize, and chromosomes segregate faithfully), but essential in Meiosis II. They could discuss this paper (that they cite) in light of their results.

- In the introduction, the authors say that compared to mammals and frogs, fish embryos are suitable for live imaging, implying that mammalian embryos are not. This is simply wrong, as mammalian embryos are highly suitable for live imaging, as shown by lots of studies that assessed in particular spindle morphogenesis in early embryos.

- The authors should at least cite and maybe comment papers from the Dumont lab (nematode meiotic divisions) showing that chromosomes can be pushed in anaphase thanks to MT nucleation between them, and from the Basto lab (doi: 10.1016/j.cub.2019.07.061) showing a switch in spindle morphology from early to late neurogenic stages, which relies on an increase in inner spindle microtubule density and stability.

Reviewer #3 (Remarks to the Author):

Kiyomitsu et al document the dynamics of mitotic spindle assembly, and requirements for the RAN small GTPase in mitotic spindle assembly, during early and late medaka embryogenesis. Using live imaging and CRISPR-generated degraon tagging for auxin-induced degradation, the authors show that in early medaka embryos RAN is required for the assembly of a dense microtubule network at the spindle midplane that is essential for accurate chromosome separation. In contrast, RAN becomes dispensable for mitosis in later stage embryos, when spindles lack the dense mid-plane microtubule network. The authors' results nicely illustrate the virtues of medaka fish as a model system, and the authors provide several interesting observations about the nature of these rapid divisions, which exhibit highly faithful chromosome separation. The data presented are of high quality, are very interesting, and strongly support the authors' conclusions as to the requirements for RAN in early versus late embryos, and the manuscript provides an impressive investigation of mitotic spindle assembly in a vertebrate embryo with large early embryonic cells. The results will be of substantial interest to a wide audience of cell and developmental biologists, and overall the results warrant publication in Nature Communication, but only after the authors address the following comments.

Major comments

1. In lines 242-245, the authors conclude that microtubules between chromosomes are sufficient for chromosome separation/anaphase. However, in Figure 4A, the chromosomes appear to move closer to their respective spindle poles during anaphase. Furthermore, the authors show plots for distance between centrosomes, between separating chromosomes, and between spindle poles in 4D. But the authors do not show this quantitative data for the two levels of nocodazole treatment. The authors should show the same quantification for the nocodazole treatments, and also quantify the distance between the each spindle pole/centrosome and its proximal chromosome set during mitosis. And the authors should discuss their interpretation of chromosomes moving closer to proximal spindle poles/centrosomes if that is indeed the case. Finally, what is the difference between centrosome/centrosome distance, and spindle pole/spindle pole distance? The authors should explain this latter point in the legend.

2. Are the dense MTs around chromosomes observed after nocodazole treatment (Figure 4E) the same MTs as the dense MTs that depend on Ran-pathway (Figure 5B and 6I)? The authors should consider combining RCC1 knockdown with nocodazole 170nM treatment to determine if the dense MTs around chromosomes are indeed dependent on RCC1.

Minor comments

1. In the abstract, would it be more appropriate to end the final sentence with "...in large and rapidly dividing early embryos" (adding "rapidly dividing")?

2. In the introduction, on page 4, lines 85-87, the authors state that because medaka embryos have meroblastic cleavages, spindle dynamics can be easily analyzed. The authors should define meroblastic for a general audience, and explain how meroblastic cleavage makes imaging of spindle assembly easier. To this reviewer, incomplete cleavage does not help the imaging, but because the early divisions all occur in a relatively uniform, one cell layer thick sheet, it is easier to view and document mitotic spindle assembly throughout the embryo. Is this the advantage the authors are referring to? If so, they should state this more explicitly.

3. On page 7, lines 129/130, the authors state the in control embryos, long term imaging led to embryos developing normally until "nearly" the hatching stage. The authors should state more clearly the terminal phenotype, at least providing the percent of embryos that fail to hatch in imaged control embryos, compared to control embryos that were not exposed to laser illumination for imaging. Similarly, on page 8, lines 151-154, the authors refer to some polyploid nuclei being observed in late stage control embryos. Were such nuclei ever observed in embryos that were not laser irradiated? And again, on page 15, lines 311-313, knockdown of RCC1 led to fully penetrant embryonic lethality; how much lethality was observed when embryos in which RCC1 was not reduced in function were imaged with laser-irradiation?

4. Lines 321 and Figure 6F, the authors state that, "phenotypic severity appeared to be correlated with the expression level of OsTIR1(F74G)." However, the authors do not quantify the degree of expression with the phenotypic severity. It would be better if the authors quantitatively documented the degree of knockdown associated with the different phenotypes.

5. Based on the images in Figure 7C, the authors state that RCC1 KD blastomeres have less intense microtubules. While the overall mCh-alpha-tubulin is lower in RCC1_KD than the control, it would be nice to have a quantitative comparison of this difference (integrated pixel intensity) or perhaps line scans comparing the two.

Response to Reviewer Comments:

(Reviewers comments are shown in *blue*.)

First, we thank all reviewers for their comments and constructive criticisms and for their appreciation of our work. We have addressed all comments below in point-by-point fashion.

Reviewer #1:

Minor concerns

1. Abstract: The central statement quoting a “gradient” is to my opinion an overstatement. We all believe that RanGTP distributes in a gradient-like fashion but this is not addressed here “activity of” “high concentration of...” or just “RanGTP” (see end of intro) would do instead. I would suggest to carefully check the wording in the manuscript throughout accordingly.

We agree. We removed “gradient” from the abstract and revised the wording throughout the manuscript accordingly.

2. p.8: “synchronous” rather than “synchronized”

We corrected this as suggested.

3. sentence lanes 182/3 is not completely logical as it compares timing “down” to more cells and timing “up” to fewer cells when saying “to reach an upper limit...”

We corrected the order of sentences.

4. lanes 206/7 and 272 MT assembly instead of nucleation

We changed the corresponding sentences accordingly.

5. lane 253: corresponds to mouse and human RanT24N

Thank you for pointing this out. We corrected the typos as suggested.

6. lane 380: clearly

We corrected this as suggested.

7. Articles „the“ / „a“ if Fig. 1A; Microtubules;

We corrected this in Fig. 1a and in the text.

8. Fig. 1B: insets may be indicated in the larger views (see 1D);

We enlarged the insets as much as possible in Fig. 1b.

9. 2G: small n.

We corrected this.

10. 3D: graph: I suggest going down to 0 in the ordinate to highlight the relatively small changes in width as compared to size

Thank you for this suggestion. We revised Fig. 3d as suggested.

11. Fig. 7 D and E: I am not a friend a having non-0 ordinates...

We corrected the ordinates in these figures, which are Fig. 9e and 9g in the revised manuscript.

12. 7H: possibly highlighting the region which is most dependent of RanGTP/RCC1 activity levels (“dense MT network...”) in the central part of the spindle would be an advantage

We appreciate this suggestion. We did so, and the result can be seen in Fig. 10.

Reviewer #2:

Descriptive part of the paper (Figures 1 to 4):

1. It would be nice to show a centrosome staining instead of inferring where they are by the tubulin signal, especially for describing the Type I and II configurations of centrosome orientation (Figure 2F), and for Figure 3A-B.

We performed immunofluorescence with anti- γ -tubulin antibody and found that the signal co-localized with punctate tubulin signals showing Type I and II configurations, respectively (Supplementary Fig. 2e). Also, the anti- γ -tubulin antibody showed punctate signals in the center of astral MTs at metaphase (Supplementary Fig. 2f), consistent with a previous report (Inoue et al., *Nature Communications* 2017).

2. The centrosomes move apart from each other in metaphase (Figure 4A). For example, centrosomes in 1-cell stage move apart between 027 and 033 min (same for 2-cell stage between 063 and 069 min, and for 4-cell stage between 096 and 099 min, see Movie S2). This is less the case later on. Typically, in Figure 3A-B, when are the linescans positioned? It could matter since spindle length is defined here as the centrosome-centrosome distance. In addition, few spindles were observed at that stage: N=5 for 1-cell stage, N=9 for 2-cell stage, N=15 for 4-cell stage. These numbers should be increased, and analyzed with appropriate statistical tests.

In Fig. 3a-b, we measured metaphase spindle length using the image one frame before anaphase onset. We mentioned this in the figure. We increased N (n=18 for 1-cell stage, n=31 for 2-cell stage, n=50 for 4-cell stage) for Fig. 3c-e, and performed statistical tests.

3. The authors say that spindle width and metaphase plate length are relatively constant until the early blastula stage (Figure 3D, S3A). This should be addresses using appropriate statistical tests.

We performed one-way ANOVA with 2-cell spindle as a control for multiple comparisons and found that spindles at the 256-cell stage show significant differences in spindle length (Fig. 3c), but not in spindle width (Fig. 3d).

4. The authors conclude that since EGFP-EB1 is localized at the spindle center (Figure 3F), it means that a specialized MT network is formed around chromosomes in early spindles. Could the authors do cold treatments and then regrow of MTs, to show that they are indeed nucleated from the chromosomes? And if this network is nucleated by Ran-GTP, what is their hypothesis regarding the fact that MTs are not nucleated earlier on the chromosomes, after NEBD? The cold assays could also test the nucleation capacity of the centrosomes, that could decrease with divisions and explain the disappearance of astral microtubules with time.

According to this suggestion, we performed the cold treatment and MT regrowth assay. Although cold-stable MTs remained around chromosomes when the temperature was shifted around metaphase (Fig. 3h), MTs were almost completely destabilized around chromosomes

when shifted just after NEBD (Fig. 4i $t=0$). After shifting the temperature up, MTs gradually appeared around the condensed chromosome mass ($t=75-150$, Supplementary Movie 3). This suggests that MTs can be nucleated and/or stabilized around chromosomes. However, astral MTs appeared to reach the chromosomal region; thus, it was unclear whether MTs nucleated from chromosomes or not.

To minimize the effects of astral MTs, we also analyzed the spindle assembly process in the presence of 170 nM nocodazole (Fig. 4h). Nocodazole treatment inhibited astral MT growth, and clearly showed an increase of tubulin intensities around chromosomes at NEBD (Fig. 4h $t=0$). We also found a similar nuclear accumulation of mCherry- γ -tubulin (Fig. 4g) and EGFP-EB1 (Fig. 4f). These data suggest MT nucleation around chromosomes at NEBD. Since these MTs are dimmer compared to astral MTs, they are difficult to detect. However, they can be detected as a slight increase in tubulin intensity around chromosome in the natural process (Fig. 4c $t=0$, Fig. 4e).

We speculate that this weak MT nucleation activity at NEBD depends on Ran-GTP, based on Fig. 6b and Fig. 8a. However, we plan to test this in a future study because we need to carefully analyze the nuclear accumulation of α -tubulin together with γ -tubulin and other MT nucleation factors such as augmin. Analyzing the nucleation capacity of the centrosomes in early and late stages would also be well beyond the scope of this study.

5. EMTB-3xGFP is used as a control, because it does not accumulate at the spindle center. Looking closely at the images (Figure 3 F-G), one could wonder if the cells are exactly at the same stage. Indeed, the metaphase plate is thinner in Figure 3G compared to Figure 3F. Could it be that the spindle in Figure 3F already initiated anaphase, and maybe started nucleating microtubules between the DNA (as it is the case in nematode oocytes), whereas the spindle in Figure 3G is in metaphase? At last, the observation that EMTB-3xGFP recognizes signals around cortical cell-membranes does not mean that medaka early embryonic spindles have unique structures that consist of MTs with different characters.

To clarify this point, we showed time-lapse images of EGFP-EB1 from NEBD to metaphase (Fig. 4f) and from metaphase to anaphase (Fig. 5f). These images show that EB1 accumulates around spindle center at metaphase and between separating chromosomes during anaphase.

In live cells, EGFP- α -tubulin showed higher intensity at the spindle center than centrosomal MTs at metaphase (Fig. 3b), and the ratio (Max value of spindle MTs/Max value of astral MTs) is >1.0 (Fig. 3f). However, after cold treatment and fixation using 4% PFA and acetone, the ratio decreased (Fig. 3f), suggesting that some populations of MTs around the spindle center are sensitive to cold treatment and fixation. Similarly, during early anaphase, EGFP- α -tubulin showed higher signals between separating chromosomes (Fig. 5a, c, $t=1-2$), but these signals were reduced after fixation (Supplementary Fig. 5a), suggesting that MTs between separating chromosomes recognized by EB1 are sensitive to fixation. Interestingly, EMTB-3xGFP showed a unique intensity profile on spindles at both metaphase and anaphase: In contrast to α -tubulin and EB1, EMTB-3xGFP did not show clear accumulation at the spindle center at metaphase (Fig. 5e, Supplemental Fig. 5b). Rather it localized homogeneously on the metaphase spindle. In

addition, EMTB-3xGFP did not accumulate between separating chromosomes during anaphase (Fig. 5e). Given that this pattern is similar to images of immunofluorescence of anti- α -tubulin antibody in fixed embryos, EMTB-3xGFP may preferentially recognize stable MTs resistant to fixation, and cannot localize on dynamic, short, polymerizing MTs. Based on these new results, we think medaka early embryonic spindles consist of subclasses of MTs with different characteristics.

6. It could be worth redoing some of the key experiments in Immunofluorescence to make sure that expression of tagged proteins does not modify their localization.

We performed immunofluorescence using anti- α -tubulin antibody. Although we think our fixation protocol destabilizes some MTs based on the new results described above. Anti- α -tubulin antibody images nicely visualized individual bundled MTs, most likely k-fibers and bridging fibers or midplane-crossing MTs (Fig. 3i, Supplementary Fig. 3f). These k-fiber-like bundled MTs showed higher intensities near chromosomes (Fig. 3i, Supplementary Fig. 3f), supporting our live imaging data (Fig. 3a-b, g).

7. Overall, quantifications are done on a low number of embryos, usually not more than 4 or 5. It should be increased, as it is difficult to do statistical tests on such small numbers.

We increased n in Fig. 2b and Fig. 3c-e (18 embryos), Fig. 5b (time-lapse movies with 1-min interval from 8 embryos), and Fig. 5I (time-lapse movies with 1-min interval from 11 embryos).

8. A lot of interesting observations are reported but not quantified nor commented, such as the Type I and II configurations of centrosome orientation, the bended spindles (could it just be that the spindle is a little bit tilted in respect with the polarity axis of the cell?), comet-like structure under oil droplets, difference in cell cycle length between inner and peripheral blastomeres, MT-like signals around cortical cell membranes, lack of anaphase elongation, centrosome-centrosome increase in distance in metaphase, transient deformed structures, absence of spindle assembly checkpoint, polyploid-like nuclei, spindle scaling with cell size after the 16-cell stage and upper limit in the fourth divisions...

Thank you for your appreciation of our observations. We commented on these points in the Supplementary Discussion. Also, we added quantification for: bent spindles in the 1st division (Supplementary Fig. 3b), size of inner and outer blastomeres at the 8-cell stage (Supplementary Fig. 2b), distance between separating chromosomes and proximal spindle pole in controls (Fig. 5b) and nocodazole-treated conditions (5I), and polyploid-like nuclei (Supplementary Fig. 2c).

Functional part of the paper (Figures 5 to 7):

9. The authors are interested in Ran-GTP, but they never show that there is a Ran-GTP gradient. The authors should use FRET probes (cf Kalab 2006) to demonstrate that there is a Ran-GTP gradient, and how this gradient evolves during time along with the switch in spindle

morphogenesis observed by the authors. Then, this gradient should be measured after OI-RanT27N injection and RCC1 depletion via auxin degron.

We understand the importance of visualizing a Ran-GTP gradient, but it is technically challenging and requires huge additional work. Thus, according to the suggestion of Reviewer #1, we removed the word 'gradient' to avoid over-statement.

10. The authors manipulate Ran-GTP levels, using OI-RanT27N, that inhibits the generation of Ran-GTP. Why OI-RanT27N would cause deformation or elongation of nuclei? Could they do the reverse, using the equivalent of the Ran Q69L, which stays locked in the GTP-bound state?

We measured the lengths of long and short axes of nuclei not only in OI-RanT27N-expressing embryos, but also in OI-RanWT and OI-RanQ72L, equivalents of the mouse and human RanQ69L (Fig. 6d). This revealed that OI-RanT27N expression causes shortening of the short axis and extension of the long axis. Similar results were obtained in RCC1-depleted early embryos (Fig. 7i), but not in later-stage embryos (Fig. 9c). In contrast, OI-RanQ72L expression did not cause significant changes in nuclear size or spindle assembly (Fig. 6d, Supplementary Fig. 6d), although OI-RanQ72L accumulates at the nuclear periphery (Supplementary Fig. 6g).

Ran functions for nucleocytoplasmic transport and lamin import, both of which are reported to regulate nuclear size in *Xenopus* (Levy and Heald, *Cell* 2010). In addition, nuclear laminae provide mechanical strength to nuclei (Mukherjee et al., *Nucleus* 2016). We speculate that RCC1-depleted nuclei cannot resist outward forces generated by centrosomes due to the reduced level of nuclear laminae, resulting in elongated nuclear shape (Fig. 6d, 7i).

11. I don't understand the rationale for doing the aphidicolin experiment, I would remove it.

We removed these data.

12. Quantifications are missing for lots of experiments, so it is hard to grasp the main phenotype following OI-RanT27N injection. How much cells were analyzed in total? How many displayed precocious detachments of a centrosome (some means a lot?), and why would a centrosome detach from the nucleus without Ran-GTP? How many displayed asymmetric spindles? How many had chromosome segregation defects, and what kind of defects (as in Figure 6G)? Does it cause embryonic lethality as for RCC1 depletion?

We added the information about quantification as follows:

65% (n=13/20) of OI- RanT27N-expressing embryos showed abnormal segregation in at least one blastomere before the 256-cell stage. Among them, 6 embryos had higher RanT27N expression and displayed phenotypes in 8-16 cell stages, including chromosome bridge (Fig. 6b, 76% n=25/33) and anaphase lagging chromosomes (12%, n=4/33). Injection of RanT27N mRNA caused embryonic lethality in all embryos (n=20), whereas Ran-WT injection showed normal development and high hatching rate (78%, n=7/9).

Intriguingly, among 17 tractable blastomeres showing cut phenotypes, 88% (n =15/17) had symmetrical nuclear shapes (Fig. 6e left t=0), but the remaining 12% (n= 2/17) showed asymmetrical nuclear shapes at NEBD (Fig. 6e right, t=0), which resulted in asymmetrical chromosome position relative to centrosomes (t=3), asymmetrical spindle structure (t=9), unequal distribution of segregating chromosomes (t=12-15), and unequal-sized daughter nuclei (t=24). In subsequent divisions, we additionally observed 3 blastomeres showing asymmetrical spindles, which correlated well with asymmetrical nuclear morphology (Supplementary Fig. 6e).

13. The authors conclude that chromosomes are required for spindle assembly, but are 2 centrosomes without chromosomes always making a spindle? I don't understand their conclusion, that Ran-independent chromatin pathways collaborate with centrosomes in spindle MT nucleation near centrosomes. Ran-GTP pathway can regulate microtubule assembly/dynamics, so without it, it is not surprising that less MTs are produced.

Two centrosomes without chromosomes cannot form a diamond-shaped spindle between them, but make a cleavage furrow (Fig. 6f, Supplementary Fig. 6f).

We removed previous sentences to avoid confusion. In this revised manuscript, we highlighted the fact that asymmetrical nuclear shape is caused by Ran-GTP inhibition and that this is well correlated with asymmetrical spindle formation and unequal chromosome segregation (Fig. 6e).

14. Why OI-RanT27N expression and RCC1 depletion do not cause the same defects, with stronger defects for RCC1 depletion (such as chromosome non-disjunction)? Could RCC1 depletion via auxin degron impair chromosomes, and not only the spindle, leading to some defects in chromosome segregation, since RCC1 is localized on the chromosomes?

We quantified the remaining fluorescence level of RCC1-mACF and found that the remaining RCC1 level is correlated with phenotypic severity (Fig. 7e). Based on this, we favor the idea that AID-mediated degradation of RCC1 can inhibit RCC1's activity more efficiently than OI-RanT27N expression; thus, it causes stronger defects. To test whether RCC1 impairs chromosomes, we are planning to do rescue experiments with an RCC1 mutant that lacks GEF activity, but possesses chromatin binding. This would be suitable for a subsequent next manuscript.

Minor comments:

15. In the introduction, the authors say that the Ran-GTP gradient is essential for acentrosomal spindle assembly in female meiosis. They should be more cautious, since one paper shows in mouse and xenopus oocytes that the Ran-GTP gradient is dispensable in Meiosis I (spindles can form and bipolarize, and chromosomes segregate faithfully), but essential in Meiosis II. They could discuss this paper (that they cite) in light of their results.

Thank you for this advice. We revised the sentence as follows:

Consistent with this model, Ran-GTP is essential for acentrosomal spindle assembly in female meiosis³¹⁻³³, with more functional significance in meiosis II in mice³¹.

16. In the introduction, the authors say that compared to mammals and frogs, fish embryos are suitable for live imaging, implying that mammalian embryos are not. This is simply wrong, as mammalian embryos are highly suitable for live imaging, as shown by lots of studies that assessed in particular spindle morphogenesis in early embryos.

We revised the sentences as follows:

Fish embryos generate large, centrosome-containing spindles^{7, 39-41}. Importantly, early divisions are planar^{2, 14, 40}. They occur synchronously in a relatively uniform, single cell-layer sheet at the animal pole, providing an ideal opportunity to observe dynamic processes of spindle assembly and positioning in all blastomeres from 1-cell to 16-cell stages.

17. The authors should at least cite and maybe comment papers from the Dumont lab (nematode meiotic divisions) showing that chromosomes can be pushed in anaphase thanks to MT nucleation between them, and from the Basto lab (doi: 10.1016/j.cub.2019.07.061) showing a switch in spindle morphology from early to late neurogenic stages, which relies on an increase in inner spindle microtubule density and stability.

Thank you for your suggestions. I discussed the contribution of TPX2 level with Vargas-Hurtado et al., from the Basto lab, and Helmke and Heald from the Heald lab. We also discussed how MTs are regulated during chromosome separation in the Supplementary Discussion by citing Laband et al., from the Dumont lab.

Reviewer #3:

Major comments

1. In lines 242-245, the authors conclude that microtubules between chromosomes are sufficient for chromosome separation/anaphase. However, in Figure 4A, the chromosomes appear to move closer to their respective spindle poles during anaphase. Furthermore, the authors show plots for distance between centrosomes, between separating chromosomes, and between spindle poles in 4D. But the authors do not show this quantitative data for the two levels of nocodazole treatment. The authors should show the same quantification for the nocodazole treatments, and also quantify the distance between the each spindle pole/centrosome and its proximal chromosome set during mitosis. And the authors should discuss their interpretation of chromosomes moving closer to proximal spindle poles/centrosomes if that is indeed the case. Finally, what is the difference between centrosome/centrosome distance, and spindle pole/spindle pole distance? The authors should explain this latter point in the legend.

First, we defined the position of chromosomes, spindle poles, and centrosomes based on the fluorescent intensity profile along the long axis of the spindle (see Fig. 3b, Fig. 5c). The definition is found in the Figure legend of Fig. 5c. In contrast to typical somatic cells, centrosomes are separated from focused spindle poles at metaphase in medaka early embryos (Fig. 3b, Fig. 5c). Centrosome-centrosome distance continuously increases during anaphase, whereas pole-pole distance increases only slightly (Fig. 5a-b).

We carefully quantified chromosome-pole distance from metaphase to anaphase using time-lapse movies with 1-min intervals (Fig. 5a-b). As the reviewer pointed out, separating chromosomes move closer to their respective spindle poles during anaphase so that the chromosome-pole distance decreases (Fig. 5b). Similarly, chromosomes move toward their spindle poles in nocodazole-treated blastomeres (Fig. 5k-l). We did not measure the centrosome-centrosome distance in nocodazole-treated blastomeres because centrosome positions became unclear.

Since our fixed and live images showed that MTs between chromosomes and spindle poles became shorter during anaphase (Fig. 5d-e), we favor the idea that k-fibers (Fig. 3j) are depolymerized during anaphase, like typical anaphase in somatic cells. In addition, considering an increase of pole-pole distance in nocodazole-treated anaphase spindles (Fig. 5k-l), sliding of anti-parallel MTs connected to sister k-fibers (bridging fibers) would also contribute to chromosome separation (Fig. 5m).

In contrast to EMTB-3xGFP (Fig. 5e), EGFP-EB1 accumulated between separating chromosomes during anaphase (Fig. 5f). Plus-ends of bridging fibers may extend to generate anti-parallel MT regions during anaphase. Alternatively, we do not exclude the possibility that polymerizing MTs generate pushing forces to separate chromosomes, as in *C.elegans* meiotic spindles.

Intriguingly, our quantification showed that separating chromosomes move faster ($\sim 3\text{-}4\ \mu\text{m}/\text{min}$ in controls and $\sim 1.6\text{-}2.5\ \mu\text{m}/\text{min}$ in nocodazole-treated blastomeres) than typical anaphase

chromosomes (~1 $\mu\text{m}/\text{min}$). Early embryos could achieve this by increasing k-fiber depolymerization activity and/or sliding of bridging fibers, or by exerting additional mechanisms in parallel. We discuss these possibilities in Supplementary Discussion.

Finally, when we reanalyzed data of nocodazole-treated cells, we noticed that nocodazole-treatment highlights tubulin accumulation in the nucleus at NEBD (Fig. 4h) and subsequent self-assembly of spindle without robust astral MTs. We appreciate your thoughtful comments.

2. Are the dense MTs around chromosomes observed after nocodazole treatment (Figure 4E) the same MTs as the dense MTs that depend on Ran-pathway (Figure 5B and 6I)? The authors should consider combining RCC1 knockdown with nocodazole 170nM treatment to determine if the dense MTs around chromosomes are indeed dependent on RCC1.

We combined RCC1 knockdown with nocodazole treatment (Fig. 8f-g). Since 170 nM nocodazole treatment did not inhibit astral MT growth efficiently after mRNA injection and 5-Ph-IAA treatment compared to normal conditions (Fig. 3g), we used 330 nM nocodazole.

Our results indicate that the dense MTs around chromosomes are partially dependent on RCC1. mCh- α -tubulin was still detected, but intensities were significantly reduced (Fig. 8f-g). This indicates that MTs around chromosomes after nocodazole treatment are partially identical to dense MTs that depend on RCC1 (Fig. 8a). Remaining MTs in RCC1-depleted blastomeres appeared to be resistant to nocodazole-treatment.

Minor comments

1. In the abstract, would it be more appropriate to end the final sentence with "...in large and rapidly dividing early embryos" (adding "rapidly dividing")?

We agree. We corrected the final sentence as suggested.

2. In the introduction, on page 4, lines 85-87, the authors state that because medaka embryos have meroblastic cleavages, spindle dynamics can be easily analyzed. The authors should define meroblastic for a general audience, and explain how meroblastic cleavage makes imaging of spindle assembly easier. To this reviewer, incomplete cleavage does not help the imaging, but because the early divisions all occur in a relatively uniform, one cell layer thick sheet, it is easier to view and document mitotic spindle assembly throughout the embryo. Is this the advantage the authors are referring to? If so, they should state this more explicitly.

Thank you for your kind and constructive criticism. We agree. We changed the corresponding sentences as shown below:

Fish embryos generate large, centrosome-containing spindles^{7, 39-41}. Importantly, early divisions are planar^{2, 14, 40}. They occur synchronously in a relatively uniform, single cell-

layer sheet at the animal pole, providing an ideal opportunity to observe dynamic processes of spindle assembly and positioning in all blastomeres from 1-cell to 16-cell stages.

3. On page 7, lines 129/130, the authors state the in control embryos, long term imaging led to embryos developing normally until “nearly” the hatching stage. The authors should state more clearly the terminal phenotype, at least providing the percent of embryos that fail to hatch in imaged control embryos, compared to control embryos that were not exposed to laser illumination for imaging. Similarly, on page 8, lines 151-154, the authors refer to some polyploid nuclei being observed in late stage control embryos. Were such nuclei ever observed in embryos that were not laser irradiated? And again, on page 15, lines 311-313, knockdown of RCC1 led to fully penetrant embryonic lethality; how much lethality was observed when embryos in which RCC1 was not reduced in function were imaged with laser-irradiation?

We corrected the sentence as shown below:

Despite long-term imaging, 82% (n=18/22) of embryos developed normally and hatched like control embryos without imaging (85%, n=28/33), suggesting very low phototoxicity of these imaging conditions.

We used live imaging data of hatched embryos for quantification (Fig. 2b-c, 2g, Fig. 3b-e, Fig. 4c-d, Fig. 5b-c). We mentioned this in the Methods.

Regarding polyploid-like nuclei, we observed them in embryos that later hatched. In addition, we confirmed that they can be seen in embryos that were not laser irradiated. These polyploid like nuclei were frequently observed on the surface of yolk, but not in the embryonic body. We discussed this in the Supplementary Discussion due to space limitations.

For RCC1 depletion in 4-cell stage embryos, we observed 28 blastomeres from 7 embryos as -OsTIR1 controls. All 7 embryos hatched, whereas all 14 embryos showed embryonic lethal phenotypes when OsTIR1 was expressed. We mentioned this in the revised manuscript.

In addition, embryos showed 100% embryonic lethality (n=0/22) when RCC1 was depleted in 8 hpi. In this case, -OsTIR1 control showed a 70% hatching rate (n=7/10). 3 embryos did not hatch, but developed normal fish-shaped bodies. We added this information in the revised manuscript.

4. Lines 321 and Figure 6F, the authors state that, “phenotypic severity appeared to be correlated with the expression level of OsTIR1(F74G).” However, the authors do not quantify the degree of expression with the phenotypic severity. It would be better if the authors quantitatively documented the degree of knockdown associated with the different phenotypes.

We quantified remaining fluorescent intensities of RCC1-mACF in the nucleus one frame before NEBD of 4-cell division and found that phenotypic severity was indeed correlated with the knockdown level of RCC1 (Fig. 7e, 7g).

5. Based on the images in Figure 7C, the authors state that RCC1 KD blastomeres have less intense microtubules. While the overall mCh- α -tubulin is lower in RCC1_KD than the control, it would be nice to have a quantitative comparison of this difference (integrated pixel intensity) or perhaps line scans comparing the two.

We quantified intensities of mCh- α -tubulin along the spindle long axis using line scan (Fig. 9d) and measured the MT density index (Minimum value around spindle center (Min)/ Max value around spindle pole (Max)) for controls and RCC1 knockdown blastomeres. This quantitative comparison showed that MT density is reduced around chromosomes in RCC1-depleted cells (Fig. 9f).

REVIEWERS' COMMENTS

Reviewer #1 (Remarks to the Author):

I am quoting from my initial review, in which I stated that I was very enthusiastic about this work ... for being technically excellent, for taking on the (from my point of view) conceptually exactly right interpretation, and, for providing a reasonable explanation as to why previous data on Ran's role in spindle formation focused on embryonic situations while being less hard to grasp in somatic cells.

The few formal issues that I had raised were certainly addressed, and I am happy to support publication of the manuscript in its present form.

Reviewer #2 (Remarks to the Author):

The points raised in the previous round of review have been satisfactorily addressed. The authors have done their best to address most of my comments and concerns, adding new data and convincing explanations in the text where needed. I fully support the publication of their work.

Reviewer #3 (Remarks to the Author):

The authors have very thoroughly addressed all reviewer comments and have added a substantial amount of new data and clarifications to their manuscript. In my opinion the manuscript is much improved and warrants publication in Nature Communications with no further revisions needed.